



# CREST-VEC: A framework towards more accurate and realistic flood simulation across scales

Zhi Li[1], Shang Gao[1], Mengye Chen[1], Jonathan J. Gourley[2,] Naoki Mizukami[3], Yang Hong[1]

[1]School of Civil Engineering and Environmental Science, University of Oklahoma, Norman, OK, 70372, USA
[2]NOAA/National Severe Storms Laboratory, Norman, OK, 73072, USA
[3]National Centers for Atmospheric Research, Boulder, CO, 80307, USA

*Correspondence to*: Yang Hong (yanghong@ou.edu)

**Abstract.** Large-scale (i.e., continental and global) hydrologic simulation is an appealing yet challenging topic for the
hydrologic community. First and foremost, model efficiency and scalability (flexibility in resolution and discretization) have
to be prioritized. Then, sufficient model accuracy and precision are required to provide useful information for water
resources applications. This study presents a hydrologic modeling framework CREST-VEC (Coupled Routing and Excess
STorage-VECtor based routing) that combines a gridded water balance model and a newly developed vector-based routing
scheme. First, in contrast to a conventional fully gridded model, this framework can significantly reduce the computational
cost by at least ten times, based on experiments at regional (0.07 sec/step vs. 0.002 sec/step) and continental scales (0.35
sec/step vs. 7.2 sec/step). This provides adequate time efficiency for generating operational ensemble streamflow forecasts
and even probabilistic estimates across scales. Second, the performance using the new vector-based routing is improved,
with the median-aggregated NSE (Nash-Sutcliffe Efficiency) score increased from -0.06 to 0.13 over the CONUS. Third,
with the lake module incorporated, the NSE score is further improved by 62.5% and the systematic bias is reduced by 36.7%.
Lastly, over 20% of the false alarms on two-year floods in the US can be mitigated with the lake module enabled, at the
expense of only missing 2.3% more events. This study demonstrated the advantages of the proposed hydrological modeling
framework that could provide a solid basis for continental and global scale water modeling at fine resolution. Furthermore,
the use of ensemble forecasts can be incorporated into this framework; and thus, optimized streamflow prediction with
quantified uncertainty information can be achieved at operational fashion for stakeholders and decision-makers.

# 1 Introduction

Flooding all over the world has affected millions of people, especially those who reside in floodplains (Li et al., 2021a;
Tellman et al., 2021). In the US, flooding, as the primary cause for billion-dollar weather disasters, costs $3.9 billion
monetary losses and 15 deaths per year over the past four decades according to the NOAA National Centers for
Environmental Information U.S. Billion-Dollar Weather and Climate Disasters (2021). In light of frequent flooding in the
30  US, several public agencies have been operating real-time flood forecasting systems, such as the NOAA NSSL (National



Severe Storms Laboratory) FLASH project (https://flash.ou.edu) and NOAA Office of Water Prediction (OWP). However, flood warnings are still either missed or unverified due to uncertainties ranging from precipitation forcing, hydrologic model structure, model parameterization, and/or hydrologic routing. As revealed by Martinaitis et al. (2017), 12.8% of flash floods in the US go completely unwarned per year, let alone falsely warned. Apart from pursuing accurate weather forecasts,
35   improving hydrologic simulation is the key to issuing flood warnings properly.

Flow routing in hydrology is the lateral transport of water on the land surface, subsurface, and in waterways (namely hydrologic compartments). It is an inseparable component in hydrologic simulation to redistribute and exchange water between compartments and also the most time-consuming part of the simulation. In a lumped hydrologic model (watershed as an integrated unit), routing can be simplified to convolution in time, such as the Unit Hydrograph (UH) or referred to
40   Impulse Response Function (IRF) (Chow, 1968). However, variable velocities over the land surface and in waterways are difficult to be physically considered. Parameterization is pragmatic, but too many parameters could lead to equifinality (Beven, 2006). In addition, only outlet streamflow can be simulated in a lumped model. A semi-distributed model was then born to resolve flow pathways using Digital Elevation Models (Quinn et al., 1991). Owing to increasing computing power, gridded hydrologic models with spatially distributed routing become feasible over large domains (Shaad, 2017). Terrain (or
45   hillslope) routing and river channel routing at grid scales can be explicitly represented in model settings with distributed solvers such as linear reservoir (Liston et al., 1994; Wang et al. 2011), kinematic wave model (Vergara et al., 2016), and diffusive wave model (de Almeida & Bates, 2013; Lighthill & Whitham, 1955; Ponce et al., 1978). More recently, vector-based routing has attracted more attention instead of raster-based routing for large-scale (i.e., continental and global extent) simulation. In theory, vector-based routing and raster-based routing differ in defining unit catchments and river networks.
50   For instance, a raster-based routing model discretizes both catchments and river networks on Cartesian coordinates, while a vector-based routing model builds upon the irregular shape of unit catchments (i.e., polygon) and river networks (i.e., polyline).

The pioneering experiment of vector-based routing can be dated back to the early 2000s, in which river network models were incorporated in emerging Geographic Information System (GIS) software (Wang et al., 2000). With the burgeoning
55   availability of global-scale hydrography datasets (e.g., HydroSHEDS and NHDPlus), vector-based routing models have been gaining considerable interest in recent years (David et al., 2011; Lehner & Grill, 2013; Mizukami et al., 2016; Paiva et al., 2011; Yamazaki et al., 2011;). Among those developments, three frameworks have become popular and stand out in the hydrologic model community. First, David et al. (2011) introduced the RAPID routing framework that is based on the Muskingum method. The RAPID has been coupled in the National Water Model operated by the NOAA OWP (Office of
60   Water Prediction) (Lin et al., 2017) and the Global Flood Awareness System (GLoFAS) developed by the ECWMF (European Center for Median-Range Weather Forecasts). Second, Yamazaki et al. (2011) developed the CaMa-Flood framework which generates flood inundation at a large scale by solving the 1D diffusive equation and spilling water over floodplains. Third, the recent development of the mizuRoute framework by Mizukami et al. (2016) offers terrain routing and multiple channel routing schemes (e.g., IRF and kinematic wave), making it more physically-based compared to RAPID





which ignores terrain routing. The mizuRoute has been used together with the hydrologic framework SUMMA (Structure for Unifying Multiple Model Alternatives) (Knoben et al., 2021) and is planned to be implemented in the CESM (Community Earth System Model). These vector-based routing models overcome several challenges for large-scale hydrologic simulations faced by raster-based routing models. First, higher model resolution in raster-based models comes at the expense of higher computational cost, which prohibits global hydrological simulations at tens or hundreds of meters. However, the vector-based routing model is much more scalable and computationally efficient, irrespective of increasing resolution. Second, river networks can be more realistically represented in a vector form. In conventional hydrologic models, the river network in a raster form has to be delineated based on a DEM as a preprocessing step. River networks generated in such a way do not always align well with natural river centerlines. For studies investigating hydrologic connectivity in particular, river grid cells in a raster form can easily become discontinuous without considering river topology. Alternatively, river networks in popular hydrography data are digitalized based on satellite optical imagery and manual inspection (Lin et al., 2020). Another weakness of raster-based routing stems from the traditional D8 flow strategy that means water in the central grid can only be permitted to flow through one of its neighboring grid cells. On the contrary, vector-based routing offers a more flexible approach (Dinf) from vector representation of river networks.

In light of the advantages of vector-based routing, this study introduces a coupled modeling framework CREST-VEC (Coupled Routing and Excess STorage with VECtor routing), which strives to facilitate real-time flood forecasting across scales. This framework seamlessly integrates the current operational flash flood forecast model structure – CREST model and the vector-based routing framework – mizuRoute. We utilize a case study to demonstrate the advantages of this coupled framework and to investigate some updates we made to improve the existing routing scheme. Three questions are posed in this regional case study: (1) Does the included subsurface routing improve model performance? (2) Can a simple natural lake simulation improve model performance in a downstream urban area? and (3) Can CREST-VEC outperform the raster-based CREST model? In the second part, we apply this framework over the continental US for a comprehensive evaluation. We ask one additional question: How many floods are falsely alarmed without considering reservoir operations? It is anticipated that findings from this work could motivate future development of large-scale hydrologic models and raise awareness on whether and how much flood forecasts by model simulations should be trusted without proper representation of lakes.

## 2. Data and methods

### 2.1 Hydrography data

In this study, we use the vectorized river network and Hydrologic Response Unit (HRU) dataset (MERIT-Hydro: Lin et al., 2019) derived from the high-accuracy Multi-Error-Removed-Improved-Terrain (MERIT) Hydro hydrography dataset (Yamazaki et al., 2019). The flowlines were created from the 90-meter DEM data (MERIT DEM), covering the full global land surface (60°S-90°N). A minimum channelization threshold of 25 km$^2$ (upstream area) was applied to restrict river channel grid cells in the MERIT Hydro dataset. The HRUs were processed along with flowlines by the TauDEM software



and trimmed with the HydroBASINS level-II boundaries. Detailed processing of the hydrography data is listed in Lin et al. (2019). This set of hydrography data has been validated against 30-year Landsat imagery (Lin et al., 2021) and empowered global reconstruction of historical streamflow (Lin et al., 2019; Yang et al., 2021). Over the CONUS, we have obtained

100    341,921 river reaches and the same amount of unit catchments for the routing component.

Lakes and reservoirs in the U.S. play a significant role in regulating streamflow (Tavakoly et al., 2021). Major river basins (e.g., Mississippi and Columbia River Basins) are highly regulated, as shown in Fig.1a and 1b, results obtained from Lehner et al. (2011). The HydroLAKES dataset provides a global catalog of lake polygons and pour points that can be easily integrated into hydrologic models (Messager et al., 2016). Over one million natural lakes and constructed reservoirs were

identified globally, with minimum surface area larger than 10 ha. Over the U.S., there are 96,874 lakes recorded in the HydroLAKES data, of which 94,865 are natural lakes without human intervention, and 1,992 (17) lakes are reservoirs (regulated lakes), as shown in Fig.1c. Of regulated lakes or reservoirs, 20.0% are primarily used for irrigation, 19.9% for hydroelectricity, 17.6% for water supply, 17.2% for flood control, 14.1% for recreation, 1.9% for navigation, 0.7 for fisheries, and 8.6% for others (Fig.1d). The total lake volume, estimated from the lake bathymetry, is a required field to our

modeling framework to approximate outflow.

[INSERT FIGURE 1 HERE]

**Figure 1. Maps of (a) percent of regulated river and (b) regulated lake volume; (c) bar plot of lake classifications; (d) pie plot of US regulated lake or reservoir function purposes.**

**2.2 Forcing data**

Forcing data is required as model inputs to drive the hydrologic model. Hourly precipitation rates are obtained from the Multi-Radar Multi-Sensor (MRMS) data, operated at the NOAA NSSL (Zhang et al., 2016). The MRMS is a state-of-the-art radar-gauge merged product, providing instantaneous rates at a 1-km spatial resolution over the CONUS and parts of southern Canada and northern Mexico. We used the one-hour accumulated and gauge-corrected rainfall product in this study for streamflow simulation. The performance and hydrologic utility of MRMS data has been corroborated in previous studies

(Li et al., 2020, 2021b). The daily temperature from the PRISM (Parameter-elevation Relationships on Independent Slopes Model) is used to simulate snow accumulation and melt (PRISM Climate Group, 2014). The PRISM team routinely collects meteorological data from meteorological stations over the U.S. and interpolates them into 4-km gridded data based on the elevation dependence (Daly et al., 2008). The potential evapotranspiration (PET) data is obtained from the USGS FEWS data port (https://earlywarning.usgs.gov/fews) at daily and 1° spatial resolution (Allen et al., 1998). Forcing data at different

spatial resolutions are re-gridded to a 1-km model resolution. All of these data are collected from the simulation period in complete calendar days from 2015 to 2019.



### 2.3 CREST model

As jointly developed by the University of Oklahoma (OU) and NASA, the CREST model has been released for a decade (Wang et al., 2011). It is a distributed hydrologic model whose primary purposes are (1) flood simulation and forecasting, (2) evaluating the hydrologic utility of satellite precipitation datasets, and (3) water resources management (Xue et al., 2013; Tang et al., 2016; Gourley et al., 2017; Chen et al., 2020; Li et al., 2021b). Owing to its relatively simple structure and computationally efficient simulation, the CREST model has been promoted by the NOAA NSSL for real-time flash flood forecasting over the continental U.S. and its territories (Gourley et al., 2017; Flamig et al., 2020). As shown in Fig. 2, the effective rain (deficit of rainfall rates and evaporation rates) reaches the land surface and is partitioned into fast runoff from urban impervious area ratio and infiltration into the soils. A Variable Infiltration Curve (VIC) model is incorporated to determine the infiltration rate (Liang et al., 1994). Surface runoff is generated when infiltration rates become higher than the maximum infiltration capacity. In the meantime, slow-flowing interflow is produced while soil water content is depleted. In the CREST model, flow routing is handled in two ways. Terrain routing and in-channel river routing are done by the kinematic wave model which simplifies the Saint-Venant equation by ignoring the acceleration and forcing terms (Vergara et al., 2017). The interflow is routed by a conceptual linear reservoir with parameterized velocity (Shen et al., 2017). We refer to the CREST model hereafter as a standalone package that couples the water balance model with gridded terrain and channel routing.

To account for snowmelt, we coupled the original CREST model with the Snow-17 model, which is part of the National Weather Service River Forecast System in the U.S. (Franz et al., 2008). The Snow-17 model is a conceptual snowmelt scheme that simulates snow accumulation and ablation based on temperature and precipitation as inputs (Anderson, 2006). Although the physics behind it is not as comprehensive as the energy balance model, Snow-17 is advantageous for having less required input data and performing "at least as good as" energy-based models (Ohmura, 2001).

### 2.4 mizuRoute

The mizuRoute river routing model, developed at the NCAR (National Center for Atmospheric Research), is a vector-based routing framework that incorporates both terrain and channel routing for large-domain river routing applications (Mizukami et al., 2016, 2021). For the terrain routing, the IRF or UH is used with parameters associated with gamma distribution to adjust the shape and scale. For the channel routing, user-defined options are IRF, kinematic wave with Lagrangian solution, and kinematic wave with Euler solution. A recent version of mizuRoute (Version 2.0.1) includes two lake routing schemes (Gharari et al., in prep; Vanderkelen et al., 2022) – one based on Döll et al. (2003) with a simple level-pool equation for natural lakes and the other more complicated one based on Hanasaki et al. (2006) which includes reservoir operation rules. These two schemes have been applied for the other global hydrologic models (e.g., WaterGAP, VIC, and CWatM) to account for regulated streamflow.





The current version of mizuRoute does not explicitly account for subsurface runoff routing over terrain, which is critical in the Great Plains and regions where streams are intermittent across a year (Salas et al., 2017). In this study, we enable an

option to turn on or off subsurface routing as defined in the model configuration file. Similar to surface runoff routing, the subsurface flow is routed using the IRF scheme but with much slower velocity and reduced magnitude.

## 2.5 CREST-VEC

The framework, CREST-VEC, and the difference compared to its precedent CREST model is shown in Fig. 2. The main difference comes from the routing process where the original CREST model routes surface flow and interflow via a

kinematic wave routing model and a conceptual linear reservoir model in a gridded manner. However, the CREST-VEC model requires area-averaged time series of surface and subsurface flow at each river reach to be separately routed downstream. The gridded outputs from the CREST model (i.e., surface runoff and subsurface runoff) are extracted and averaged over each unit catchments or HRU using the newly developed Python package EASYMORE (EArth SYstem Modeling REmapper), publicly available from https://github.com/ShervanGharari/EASYMORE.

We use the IRF scheme for both terrain routing and channel routing in this study and activate the lake model with the Döll et al. (2003) lake model. The parameters for IRF routing are based on default values provided by Mizukami et al. (2016), and the lake parameters, such as the outflow coefficient $a$ and exponent $b$ of eq.1, are based on suggested values in Döll et al. (2003) and Gharari et al. (in prep.). For lakes that have monitored storage provided by the US Geological Survey (USGS), we directly insert storage time series into the model. As reservoir operation is not considered in this study, we exclude

observed streamflow that is regulated by reservoirs and regulated lakes, as shown in Fig. 1c. So, only results from natural lakes, which account for 98% of US lakes or reservoirs, are considered valid for statistical comparison. To initialize model states, especially for initial lake volumes, we warm up the CREST-VEC model from 1948 to 2014 using the GLDAS forcing (Global Land Data Assimilation System) at a daily time step.

$$Q_{out} = a \times S_f \times (S_f/S_{f,max})^b, \tag{1}$$

where $a$ and $b$ are the outflow coefficient (1/day) and exponent, respectively; $S_f$ is the actual lake storage (m³); $S_{f,max}$ is the maximum lake storage (m³).

[INSERT FIGURE 2 HERE]

**Figure 2. Schematic view of the *CREST-VEC* framework. The red arrow highlights the newly added subsurface routing option to the original mizuRoute framework.**

## 185  3. Results

### 3.1 Case study: Houston region

As mentioned in the objectives of this study, we first conduct a case study analysis to assess the relative contributions of subsurface flow routing and lake routing to streamflow simulation based on the CREST-VEC framework. The original





CREST model is used as a benchmark. We chose the Houston region (Fig. 3a) because there are two large natural lakes -
Lake Barker and Lake Addicks that impact hydrologic simulations (Fig. 1a). For the CREST model with gridded routing, we
calibrate the model using the DREAM (Differentiable Evolution Adaptive Metropolis) optimization (Vrugt et a., 2009) from
2016-06-01 to 2017-06-01 at an hourly time step and perform evaluation from 2017-06-01 to 2020-01-01. We run the
CREST model at three spatial resolutions: 1 km, 250 m, and 90 m. To be comparable with CREST-VEC simulations, whose
hydrography data is built upon a 90-meter resolution DEM, we only use CREST model results at 90 meters for statistical
comparisons and use the results at 1 km and 250 meters to assess computational efficiency. The evaluation metrics shown in
Fig. 3c are based on the evaluation period. The river flows from 22 stream gauges are curated from the USGS.

Figure 3b shows the computational cost (elapsed time at seconds per step) for a series of model configurations. All the tests
were run on a single core Intel i7-6700K CPU (4.00 GHz). The grid-based CREST model costs 0.01, 0.08, and 0.12 seconds
per step at 1-km, 250-m, and 90-m resolutions, respectively. However, the CREST-VEC model can reduce this to
approximately 0.002 seconds per step, regardless of grid resolutions from forcing data. There is little difference among the
three scenarios (i.e., CREST-VEC, CREST-VEC+subq: CREST-VEC plus subsurface routing, and CREST-VEC+subq+lake:
CREST-VEC with subsurface routing and lake routing). Relatively speaking, CREST-VEC can speed up the current
operational CREST model at 1-km by 10x, let alone at finer resolutions.

Regarding model skill, the CREST model and CREST-VEC achieve similar median NSE (Fig. 3c) based on observations
from 22 stream gauges, even though the CREST model takes advantage of automatic calibration. CREST-VEC and CREST-
VEC+subq overestimate flows downstream of two natural lakes, resulting in poor scores. But after incorporating lake routing
schemes, the CREST-VEC+subq+lake model achieves not only better median scores but also less spread (quantified by the
interquartile range). Notably, both CREST-VEC+subq and CREST-VEC+subq+lake have positive NSE values and smaller
uncertainty ranges, primarily owing to included subsurface routing. The time series in Fig. 4 highlights the model
performance at three stream gauges affected by upstream lakes. The CREST-VEC overestimates streamflow by a
considerable amount (i.e., three times higher than observation in Hurricane Harvey), resulting in low NSE scores – 0.11,
0.16, and 0.18, respectively. With lake routing considered in the CREST-VEC+subq+lake, the simulated streamflow aligns
well with observations, achieving NSE scores of 0.61, 0.65, and 0.64, respectively. Although the CREST model captures
streamflow magnitude after calibration with the NSE scores – 0.37, 0.52, and 0.54, the peak timing is at least one-day
delayed for Hurricane Harvey. In summary, the advantages for the general CREST-VEC framework against the gridded
CREST model are threefold: (1) improve computational efficiency by at least ten times, (2) improve overall model skill, (3)
reduce uncertainty ranges.

[INSERT FIGURE 3 HERE]

**Figure 3. (a) Map of the study area (Houston region) showing river networks and water bodies; (b) Computation time per step for**
**CREST at three resolutions and CREST-VEC model at four configurations on the x-axis; (c) Nash-Sutcliffe efficiency values for**
**CREST and CREST-VEC model.**

[INSERT FIGURE 4 HERE]





**Figure 4. Performance of models downstream of two lakes. The Nash- Sutcliffe Efficiency coefficients are obtained from the CREST-VEC model with lake routing and subsurface routing. Three plots of time series of stream gauges (from upstream to**
225 **downstream: 08073500, 08073600, 08074000) are pointed aside by the map, and the Hurricane Harvey event is highlighted in red box and insets.**

### 3.2 CONUS simulation

Moving towards continental-scale hydrologic simulation, the CREST-VEC model excels at improving computational costs, leaving room for quantifying uncertainties from forcing, model structure, and parameters in real-time. For instance,
ensemble forecasts can be generated by encompassing multi-forcing, multi-model structures, and probabilistic distributions in model parameters or post-processing such as GLUE (Generalized Likelihood Uncertainty Estimation) (Beven & Freer, 2001; Beven, 2006). The following question is whether and how much the new lake routing improves a continental simulation. To answer this question, we simulate CREST-VEC with and without lake routing over the CONUS from 2017-06-01 to 2020-01-01 at an hourly time step. Notably, subsurface routing is activated for both models with and without lake
routing, so we expect the difference in results to be primarily due to lake simulation. Streamflow data from 5,350 stream gauges in the same period are collected and used for model verification.

A prior study validated the grid-based CREST model over the CONUS with the same model configuration (Flamig et al., 2020). Compared to their results in Table 1, this new framework – CREST-VEC certainly improves streamflow simulation not only via a higher resolution (from 1-km to 90-m) but with faster computational speed (7.2 sec/step to 0.37 sec/step) and
240 median NSE scores increasing from -0.06 (gridded) to 0.08 (no lake) and 0.13 (lake). The fraction of gauges with positive NSE scores has been improved from 41.8% (gridded CREST) to 50.6% (CREST-VEC without lake) and to 55.7% (CREST-VEC with lake). However, the CREST-VEC results are more biased than gridded CREST, partly due to the systematic overestimation of streamflow by the IRF routing scheme in the CREST-VEC. The difference would be primarily attributed to the different routing processes, as CREST permits leakage in the interflow reservoir, thereby leading to lower positive bias.
This also points out the need in future works to conduct a continental-scale model optimization (especially for surface water balance), which has been a long-standing scientific challenge. Results with lake simulation have reduced BIAS from 30% to 19%, as part of the water is being held in the lake. The CC (Correlation Coefficient), however, does not vary much between scenarios with and without lake simulation, as shown in Fig. 5. One of the reasons is that the CREST-VEC model does not simulate regulated lakes or reservoirs which have strong control of streamflow time shifts. Notably, the IQRs (interquartile
ranges) of NSE and BIAS for lake simulation are lower than without lake, meaning that this method particularly boosts scores at gauge locations that had poor performance previously.

Figure 6 depicts the spatial map of model skill (with lake) and its difference between scenarios with and without lake simulation. CREST-VEC with lake module in regions like the West Coast and Upper Mississippi River Basin have relatively good performance (NSE>0.4), yet over the Great Plains and East Coast, the model bias is high (BIAS>1), yielding low NSE
scores. Similar issues are found in the literature with other models (Clark et al., 2008; Konben et al., 2020; Lin et al., 2019; Mizukami et al., 2017; Newman et al., 2015; Salas et al., 2017; Knoben et al., 2020; Yang et al., 2021; Tijerina et al., 2021).



Taking the Great Plains as an example (highlighted box in Fig. 6c), we plot the annual surface runoff simulated by CREST-VEC, compared to the public dataset GRFR (Global Reach-level Flood Reanalysis) in Fig. S1. The runoff in GRFR is simulated by the VIC model and undergoes stringent bias correction against observations via the discrete quantile mapping technique (Yang et al., 2021; Lin et al., 2019). There is a 116.3% higher surface runoff by the CREST-VEC than theirs, partly explaining the high BIAS and low NSE scores in such region. We suspect the singular bulk soil layer represented in the CREST model yields such systematic differences.

However, even when accounting for multiple hydrologic model structures, performance in this region is still ranked as one of the poorest (Clark et al., 2008; Knoben et al., 2020). For example, Knoben et al. (2020) analyzed 36 hydrologic models over the U.S., in which the maximum KGE scores out of those models are lower than 0.5 over the Great Plains. Despite a common problem, research has seldom been done to characterize the poor model performance.

As shown in Fig. 7, we correlate the NSE scores with 33 environmental indices stemming from the U.S. CAMELS dataset basin attributes. Model results (i.e., *NSE_lake*) are positively correlated with high flows (*q95*), mean flows (q_*mean*), runoff ratios (*runoff_ratio*), and wet regions (*p_mean*) with statistical significance. It highlights the strength of our model for flood simulation (high flow) as its primary function. On the contrary, model performance degrades with aridity, which is widely recognized as a long-standing challenge for hydrologic simulations among other studies (Clark et al., 2008; Knoben et al., 2020; Lin et al., 2019). Increases in high or low precipitation frequency (*high_prec_freq* or *low_prec_freq*), dry periods (*low_prec_dur*), precipitation seasonality (*p_seasonality*) lower NSE scores. Besides, models tend to perform poorly for intermittent rivers, as indicated by the frequency of days with 0 flow (*zero_q_freq*), which are prevalent stream characteristics in the Great Plains (Messager et al., 2021). Other features such as the fraction of surface water in the catchment (*water_fract*), mean potential evaporation (*pet_mean*), soil depth (*soil_depth_statsgo*) are also associated with negative correlation with model performance.

Precipitation has a strong seasonal cycle that peaks during summer in the Great Plains. The frequency and duration of extreme precipitation are both high, driven by episodic mesoscale convective systems. Soil depths in the Great Plains average around one meter (Fig. 8a), close to the maximum depth (1.5 meters) over the CONUS, and the CREST model performs poorly for these deeper soils. Moreover, playas, small and rain-fed lakes that are prominent in the Great Plains, play an important role in retaining water overland and altering streamflow behaviors (Hay et al., 2016; Solvik et al., 2021).

Figure 8b shows an example of poor-performing gauges in the Southeast. Analogous to the Great Plains, the soil depths in the Southeast are considerably high (1.5 meters), leaving CREST model simulations problematic. Evapotranspiration (ET) in the Southeast is also one of the highest among the US climate divisions due to abundant precipitation, permeable soils, dense vegetation, and substantial soil radiation. Because the CREST model does not account for transpiration from vegetation nor solve the energy balance explicitly, the simulated evaporation rates may be lower than actual evaporation rates, resulting in higher effective rainfall and thus positive bias of streamflow. Therefore, the missing hydrologic processes such as transpiration and infiltration-excess process in the Southeast are likely the causations of lower NSE scores.





**Table 1. Statistical comparison of model performance over the continental U.S. Bolded numbers indicate the best metrics among the three model configurations.**

| Metrics | Gridded CREST (Flamig et al., 2020) | CREST-VEC (w/o lake) | CREST-VEC (w/ lake) |
|---|---|---|---|
| Simulation resolution | 1 km | **90 m** | **90 m** |
| Computational Speed (sec/step) | 7.2 | **0.35** | 0.37 |
| Max NSE | 0.71 | **0.87** | **0.87** |
| Median NSE | -0.06 | 0.08 | **0.13** |
| % gauges NSE>0 | 41.8 % | 50.6 % | **55.7 %** |
| Max CC | **1.0** | 0.96 | 0.96 |
| Median CC | 0.40 | **0.67** | **0.67** |
| Median bias | **9%** | 30% | 19% |

[INSERT FIGURE 5 HERE]

Figure 5. Boxplot of model performance comparing results with lake routing and without it.

[INSERT FIGURE 6 HERE]

**Figure 6. Spatial map of model performance with the lake (left column) and the difference between with and without lake simulation (right column). (a): NSE scores; (b) NSE differences (results with lake minus results without lake); (c) BIAS; (d) BIAS difference; (c) Correlation Coefficient (CC); (f) CC difference. The blue box in (c) highlights the region where high positive BIAS is present.**

[INSERT FIGURE 7 HERE]

**Figure 7. Bar plot of Spearman Correlation between 34 environmental indices (including NSE itself) from the U.S. CAMELS dataset and NSE scores simulated by CREST-VEC. Asterisk (*) signifies the statistically significant correlation between represented index with NSE (p-value < 0.05); Two asterisks (**) signify very significant correlation (p-value <0.01).**

[INSERT FIGURE 8 HERE]

**Figure 8. Boxplot of catchment attributes from CAMELS dataset in poorly performed regions and overlaid with CAMELS**
**catchments: (a) the Great Plains and (b) the Southeast. The gray shaded area represents the value range for each feature across all CAMELS gauges.**

**3.3 How likely are floods falsely detected?**

In this section, we shift gears to explore how likely US floods are falsely alarmed if no lake simulations are included. We selected 283 gauges that are downstream of natural lakes (Fig. 9), with most of them located in the middle and eastern US.
The hourly time series of streamflow of those gauges are compared against advised flood thresholds (2-year flooding) provided by the US Geological Survey. They fit a log-Pearson III type distribution to the annual maxima streamflow from long-term records and extract values with given flood frequency. Following a similar approach as in Yang et al. (2021),





consecutive yet independent events have to be two days apart from one another. From there, we calculated the Probability of Detection (POD), False Alarm Ratio (FAR), and Critical Success Index (CSI) based on the contingency table.

As expected, median FAR is reduced from 0.63 (without lake simulation) to 0.50 (with lake), a reduction of 20.6%, resulting in a slightly higher CSI of 0.36 than that of 0.31 for no lake simulation (Fig. 9a). However, such reduction comes at the expense of missing some flood events, as shown by the lower POD values (0.85 for lake and 0.87 for no lake). As most studies focus on flood detection (particularly targeting high POD), they inevitably arrive at more falsely detected floods. Too many false alarms could make people disregard the warnings, despite a real threat, causing the "cry wolf" effect.

Maps in Fig. 9b display the distributions of flood detectability with lake simulation and its improvements compared to results without lake simulation. High POD and FAR values co-exist in the Great Plains, where the model simulates considerably higher streamflow values than observations. Moderate FAR values are found near the Florida panhandle and parts of Georgia. Lower FAR values are found in the Midwest and West Coast. Compared to results without the lake, FAR values are reduced reasonably over the East Coast, Midwest, Gulf Coast, and West Coast, although POD values remain

relatively unchanged or even decreased. Overall, we observe an above 10% increase in CSI in the areas with high FAR values.

Five local cases are shown in Fig. 10, which depicts the river topology and time series of hourly streamflow. One can infer that these lakes are not heavily regulated from recorded streamflow time series, therefore showing the effectiveness of our model. In Fig. 10a, the simulated streamflow without lake is heavily overestimated, peaking at 1200 cms in the year 2017,

whereas the actual flow rate is around 400 cms. The scenario with lake simulation, however, produces a magnitude much closer to the observation. Due to decreased systematic bias, the lake scenario boosts the NSE score from -0.2 to 0.5. There is also an 8% less chance to issue false alarms than the model without lake simulation. Figure 10b shows a case where FAR is reduced from 0.70 to 0.17, a reduction rate of 75.7%. The flood detectability, i.e., CSI, is greatly improved from 0.29 to 0.57. Figure 10c exemplifies a case with all improved metrics (i.e., NSE, POD, FAR, and CSI). All these three cases in Figs. 10a-c

are located along the St. Johns River, in which we expect a systematic improvement along this river after incorporating the lake simulation. Figures 10d and 10e display more common cases where a reduction of FAR comes at the expense of reducing POD (i.e., flood detection), almost at the same pace. Consequently, the CSI values for cases with and without lake simulation are rather close, especially for Fig. 10e, where a lower CSI value is present for the lake simulation.

[INSERT FIGURE 9 HERE]

**Figure 9. Flood detection performance comparing lake and no lake simulation. (a) Similar to Figure 5, but for flood detectability; (b) Similar to Figure 6, but for flood detectability.**

[INSERT FIGURE 10 HERE]

**Figure 10. Five case examples of streamflow time series at gauges downstream of lakes: (a) St. Johns River near Sanford, FL; (b) St. Johns River near Cocoa, FL; (c) St. Johns River near De Land, FL; (d) Big Muddy River at Plumfield, IL; (e) Mississippi River**
**at Clinton, IA.**





## 4. Discussion

### 4.1 Vector vs. Raster routing

In this study, we compare the advantages of vector-based routing with respect to conventional raster-based routing in three
aspects: (1) model efficiency, (2) model accuracy, and (3) model uncertainty. Overall, the vector-based routing shows great promise, as it speeds up the model by at least ten times, compared to grid-based routing, for both the regional simulation (0.07 sec/step vs. 0.002 sec/step) and the CONUS simulation (0.35 sec/step vs. 7.2 sec/step). In terms of results against observations, the CONUS-wide performance is improved with a markedly reduced uncertainty range. However, the variable river reach lengths (from hundreds of meters to tens of kilometers) in large-scale simulation pose challenges for estimating
routing parameters such as the time and shape parameters in a unit hydrograph. Second, most land surface models are still grid-based, making a type mismatch (grid-based land surface model vs. vector-based routing model) (Lehner & Grill, 2013). To integrate the two, we need a processing step by mapping surface and subsurface runoff onto representative HRUs. Different aggregation strategies are present and subject to the primary purpose of interest. At present, there is an ongoing effort to seamlessly integrate these two processes together (Gharari et al., 2020). Third, the many-to-one river network is
established but not for one-to-many, meaning river bifurcation is challenging to represent and tackle (Yamazaki et al., 2014). Raster-based routing, on the other hand, comes at the resolution of the input DEM data albeit at a slower computational speed. Having matured over the years, most raster-based routing models are seamlessly integrated with water balance models so that model can be set up at minimum effort by a modeler. To address model efficiency, the developments of the sub-grid routing model enable more flexibility, such as lower resolution for the water balance model and higher resolution for the
routing model (Clark et al., 2015b; Getirana et al., 2012; Li et al., 2013; Shaard, 2017). Mostly derived from hydrodynamic models, the concept of "raster" can be extended to "grid" because of the emerging unstructured grids such as triangles, curvilinear, hexagons, etc. These flexible grid types mimic the real flow directions and reduce computational costs. However, they are yet to be accepted/applied by the hydrologic model community.

### 4.2 Room for improving large-scale hydrologic simulation

Large-scale hydrologic simulation is still a long-standing challenge for the hydrologic community, especially with debates on developing a "one-model-fits-all" structure or a "malleable" structure (Burek et al., 2020; Clark et al., 2015a; Fenicia et al., 2011; Savenije, 2008). We doubt that the "one-model-fits-all" philosophy is currently applicable to hydrology, in spite of some attempts to use deep learning or machine learning (ML) models that do not require a specific structure (Feng et al., 2020). The CREST model, in our study, systematically overestimates surface runoff over the Great Plains and Southeast, a
result of some misrepresented or missing processes, yet excels in flash flood simulation. Diverse hydrologic model structures, on the other hand, hope to overcome individual limitations and offer joint benefits (Horton et al., 2021). We, therefore, promote the "malleable" model structure from the efficiency point of view - a flexible structure disables redundant hydrologic processes. For instance, in tropical catchments where snowmelt hardly occurs, there is no need to activate snow



accumulation and snowmelt modules. In essence, we advocate a modular-based framework assembling different kinds of
380 model structures to serve specific needs. Then, the central question becomes: How do we adapt the model to variable
catchment processes? In such a context, intercomparisons and discussions of different hydrologic models in varying
catchment processes become particularly valuable (Clark et al., 2015a; Knoben et al., 2020; Tijerina et al., 2021). Notably,
simply relying on the NSE or KGE score to assess the model performance can be misguiding (Clark et al., 2021).

Hydrologic calibration is powerful to boost model accuracy, yet large-scale models oftentimes suffer from the complexity
that impedes credible model calibration. First, model calibration is still perceived as a patchwork solution for improving
model results. Second, how to calibrate the model in a computationally efficient way is more of an "art" than science,
requiring reflection, imagination, creativity, and inspiration. Parameter regionalization is one example that groups
parameters that share close hydrologic proximity such as soil texture, geology, climatology, etc. (Mizukami et al., 2016;
Samaniego et al., 2010, 2017). A caveat for model calibration is that most models consider hydrologic parameters under the
390 assumption of stationarity, which could be violated for climatological studies (Montanari et al., 2013).

River routing schemes and their parameters can affect streamflow simulations especially at fine time scale such as sub-daily
(Mizukami et al., 2021). Our current study used IRF scheme in which impulse response function is derived from diffusive
wave equation (see Lohman et al., 1996; Mizukami et al., 2016) and includes two parameters – diffusivity and celerity.
These parameters need to be exposed to calibration in addition to the hydrologic model parameters. Furthermore, to fully
understand routing model impact on streamflow simulations, it is necessary to consider other routing schemes including
diffusive wave as well as kinematic wave, which may be suited for flood forecasting.

While there is a proliferation of evidence that human activities are reshaping the hydrologic system, the two-way feedback
between social systems and catchment signatures is not well recognized in modern hydrologic models (Shadd, 2017). A
recent study exemplifies that incorporating water use, abstraction, and discharge data from human society improves model
simulation over the UK (Rameshwaran et al., 2022). Simulating natural lakes in a hydrologic system is only the first step
towards the goal. We intend to, in the future, adopt or infer reservoir operation rules to the CREST-VEC system via a hybrid
process-based and ML approach.

Lastly, the computational costs for large-scale simulation can be optimized from accelerated hardware (multi-core CPUs and
GPUs) once codes are parallelized and scalable. Advances in Reduced Order Modeling (ROM), a surrogate model which
develops a parsimonious solution to replace the computationally intensive part, hold promise to reduce costs (Clark et al.,
2015b). For instance, to integrate reservoir simulation into the CREST-VEC system, we can build an offline ML model
which is promising in mimicking human decisions (Yang et al., 2021) and plug it into the system.

### 4.3 How to operate flood forecasting with regulated flow?

Results in this study demonstrate a dilemma in which the model with a lake module reduces false alarms but at the cost of
410 more missed flood events compared to the one without a lake module. Although the combined metric CSI has a certain
degree of improvement, this leaves a question – should we reduce a large number of false alarms at the expense of missing a



small number of real events? Before discussing this point, we acknowledge that the current lake routing process is simple and imperfect, and improvement in this process possibly leads to an optimal situation where both false alarms and misses can be improved. However, in most situations, tradeoffs exist in hydrologic predictions. Ideally, this should involve

multidisciplinary sectors (e.g., engineers, economists, policy-makers, and urban planners) to advise flood warnings. From an engineer's point of view, a good strategy would be running both simulations with and without the lake module concurrently and making the "without lake" results the worst-case scenario. Since the CREST-VEC model has the advantage of efficiency, running two scenarios is totally feasible. A decision-maker can be trained to assess the situation – results from two scenarios disagree – from the perspective of flood severity and consequences.

**5. Conclusion**

This study compares a conventional raster-based routing scheme with the emerging vector-based routing approach in hydrologic models for a regional case and continental simulations. From the continental run, we demonstrate the improvement in streamflow simulation after incorporating the lake storage and release module. Last but not least, flood-related false alarms can be greatly reduced by including the lake module. The following points summarize the primary

findings of the study:

1. Vector-based routing can accelerate continental-scale hydrologic simulation by up to ten times, compared to a grid-based routing, for both a regional case (0.07 sec/step) and a continental case (0.002 sec/step). This leaves adequate room for generating ensemble predictions with variable forcing, parameters, and/or model structures. Furthermore, it improves streamflow simulation from -0.06 to 0.13, according to the aggregated median NSE values.

2. A newly developed lake model increases the NSE score by 62.5% and reduces systematic BIAS by 36.7% for the continental simulation.

3. Flood false alarm ratios can be mitigated by 20.6% after enabling the lake module at the expense of missing 2.3% more floods at a continental scale.

We recommend the use of ensemble simulations stemming from different model structures to overcome and adapt to varying

catchment processes. Optimized streamflow prediction with quantified uncertainty information can be achieved at operational manner for stakeholders and decision-makers. Future studies can fully investigate the limitation and uncertainty of different forcing, parameters, and/or model structures to catchment signatures such as climatology, dominant hydrologic processes, lithology, etc. Vector-based routing, in such context, can enable fair comparison by excluding the effect of different routing schemes while focusing on discrepancies in water balance models alone. For future work, we hope to have

the best possible model-simulated streamflow product in the US, fused by multi-model structures and observations. Another direction is to improve current lake and reservoir outflow simulation with a hybrid model – process-based and ML-based.



**Acknowledgement**

The first author is sponsored by the University of Oklahoma Hydrology and Water Security (HWS) program (https://www.ouhydrologyonline.com/) and Graduate College Hoving Fellowship.

**Code and data availability**

The CREST-VEC model code is publicly available in Zenodo: https://doi.org/10.5281/zenodo.6305817 (Li, 2022) (last access: February 28, 2022). The MRMS radar-based rainfall data is archived at the Iowa State University https://mesonet.agron.iastate.edu/archive/ (last access: July 3, 2021). The daily temperature data PRISM is accessed from the Oregon State University: https://prism.oregonstate.edu (last access: September 16, 2020). The hydrography MERIT-Hydro

river network data (ESRI shapefile) available at https://www.reachhydro.org/home/params/merit-basins (last access: August 10, 2021). The HydroLAKES Version 1.0 global lake data is downloaded at https://www.hydrosheds.org/page/hydrolakes (last access: August 11, 2021). The continental hourly streamflow data is archived by the US Geological Service at https://waterdata.usgs.gov/nwis/rt (last access: July 10, 2021). The script to bulk download the streamflow data is found at https://github.com/heigeo/climata.

**Author contributions**

Z.L., Y.H., and J.J.G. conceived this study; Z.L. and N.K. implemented the model code and simulated the model in this study; Z.L., S.G., and M.C. designed the experiments and analyzed the data; Z.L. wrote the original draft; All co-authors contributed to reviewing and revising the manuscript.

**Competing interests**

The authors declare that they have no conflict of interest.

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





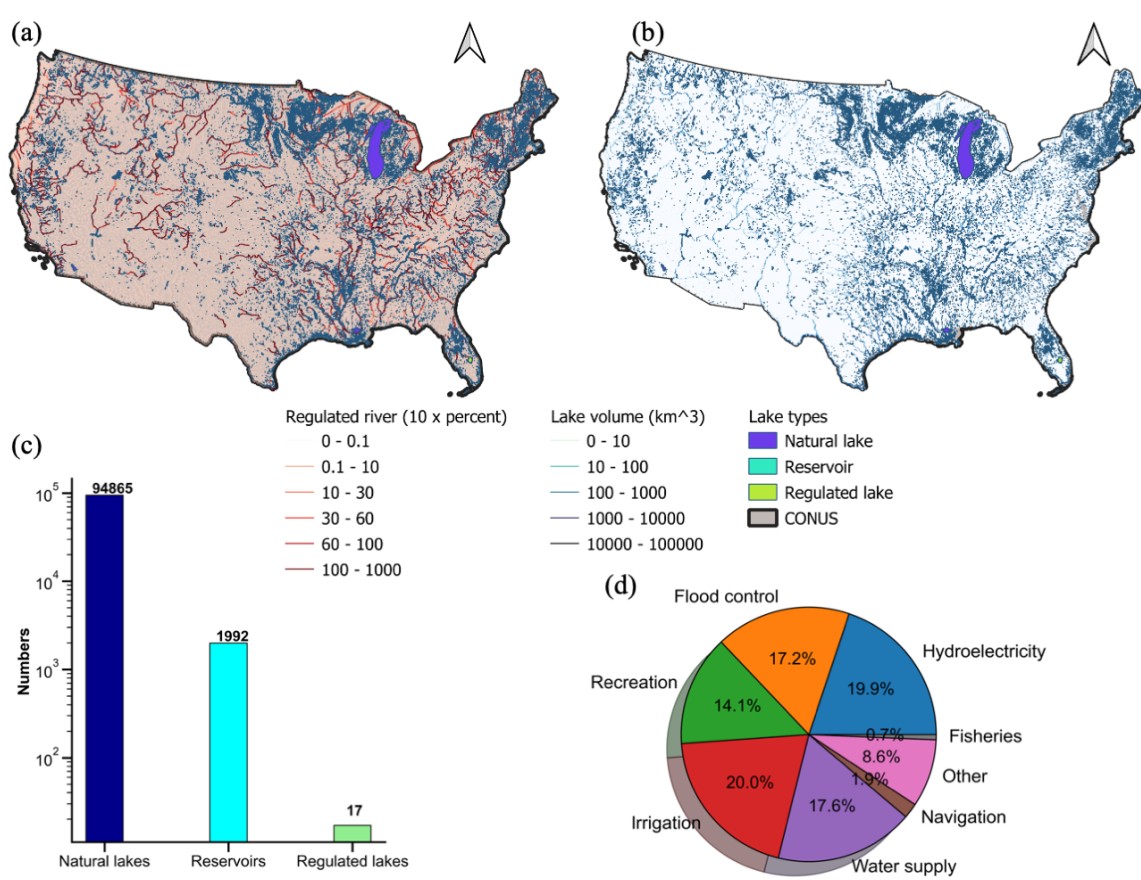

**Figure 1. Maps of (a) percent of regulated river and (b) regulated lake volume; (c) bar plot of lake classifications; (d) pie plot of US regulated lake or reservoir function purposes.**



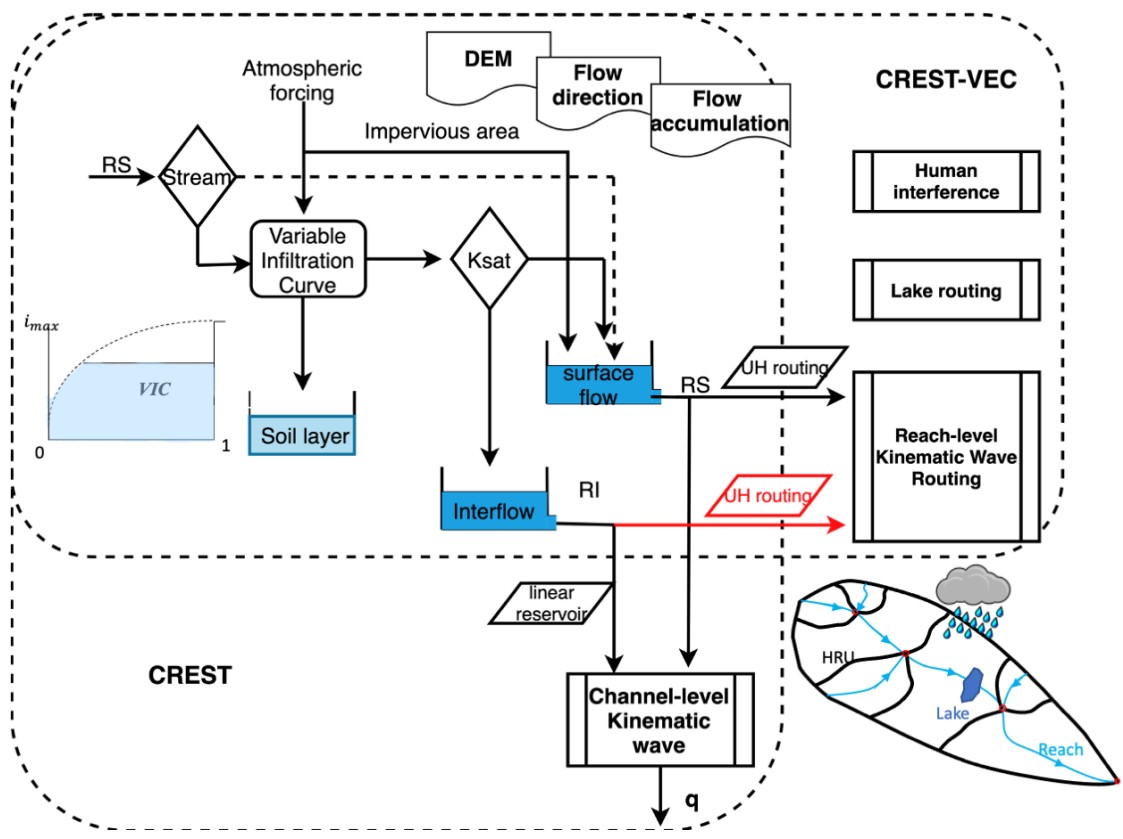

**Figure 2.** Schematic view of the CREST-VEC framework. The red arrow highlights the newly added subsurface routing option to the original mizuRoute framework.





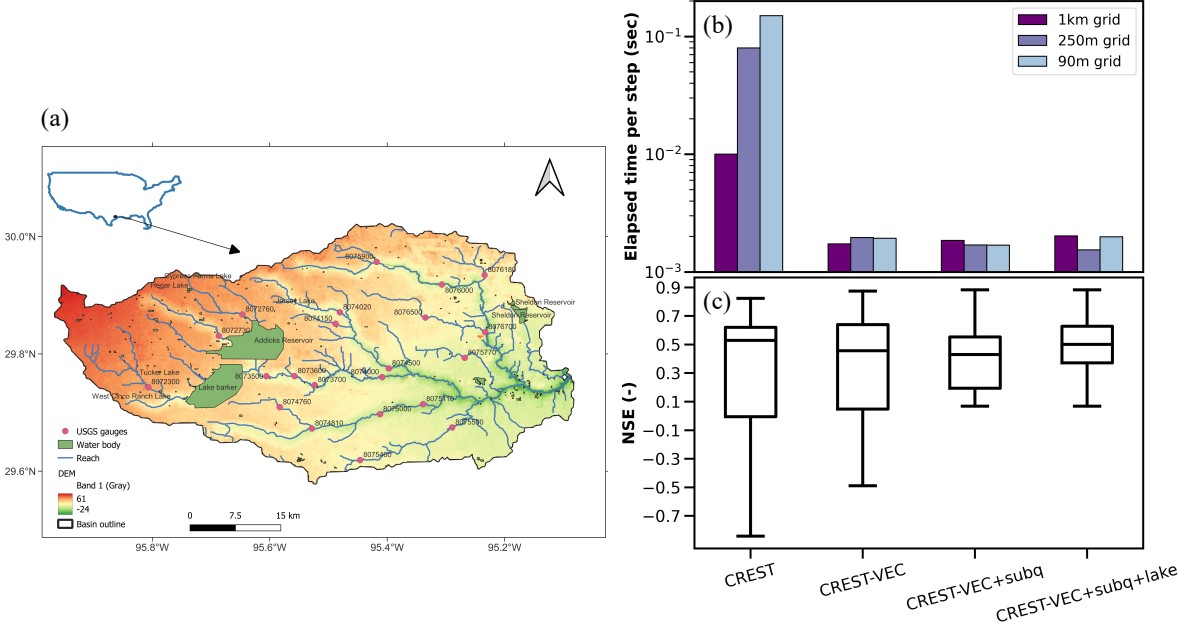

**Figure 3. (a) Map of the study area (Houston region) showing river networks and water bodies; (b) Computation time per step for CREST at three resolutions and CREST-VEC model at four configurations on the x-axis; (c) Nash-Sutcliffe efficiency values for CREST and CREST-VEC model.**



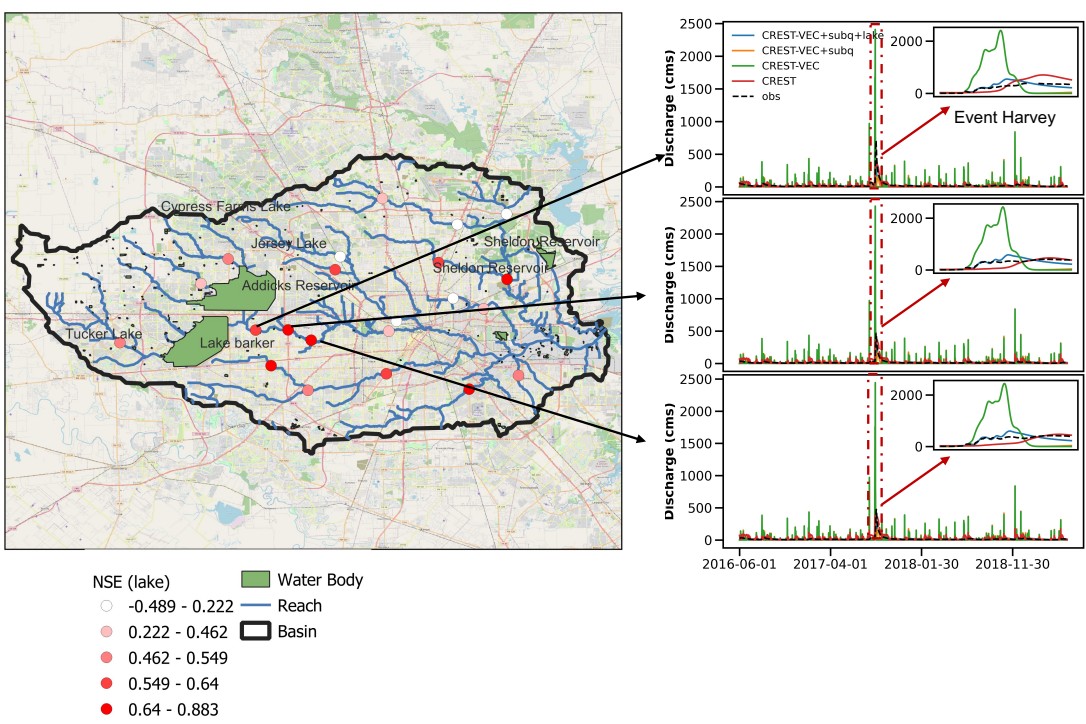

**Figure 4. Performance of models downstream of two lakes. The Nash- Sutcliffe Efficiency coefficients are obtained from the**
**CREST-VEC model with lake routing and subsurface routing. Three plots of time series of stream gauges (from upstream to**
**downstream: 08073500, 08073600, 08074000) are pointed aside by the map, and the Hurricane Harvey event is highlighted in red**
**box and insets.**





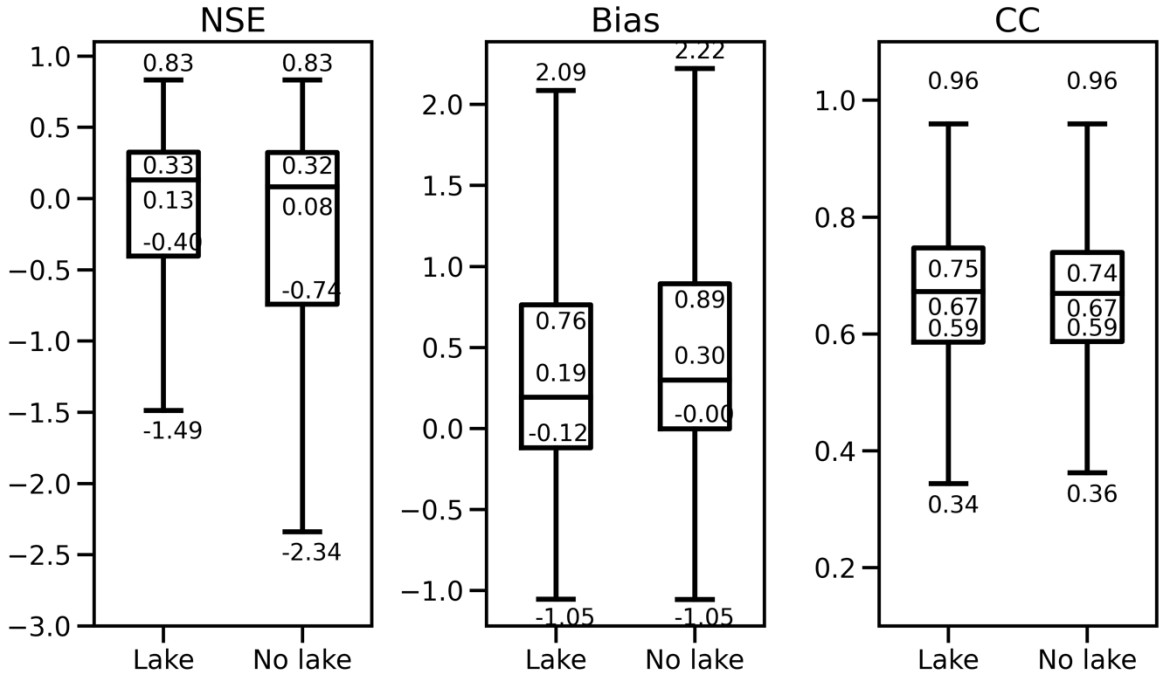

**Figure 5. Boxplot of model performance comparing results with lake routing and without it.**



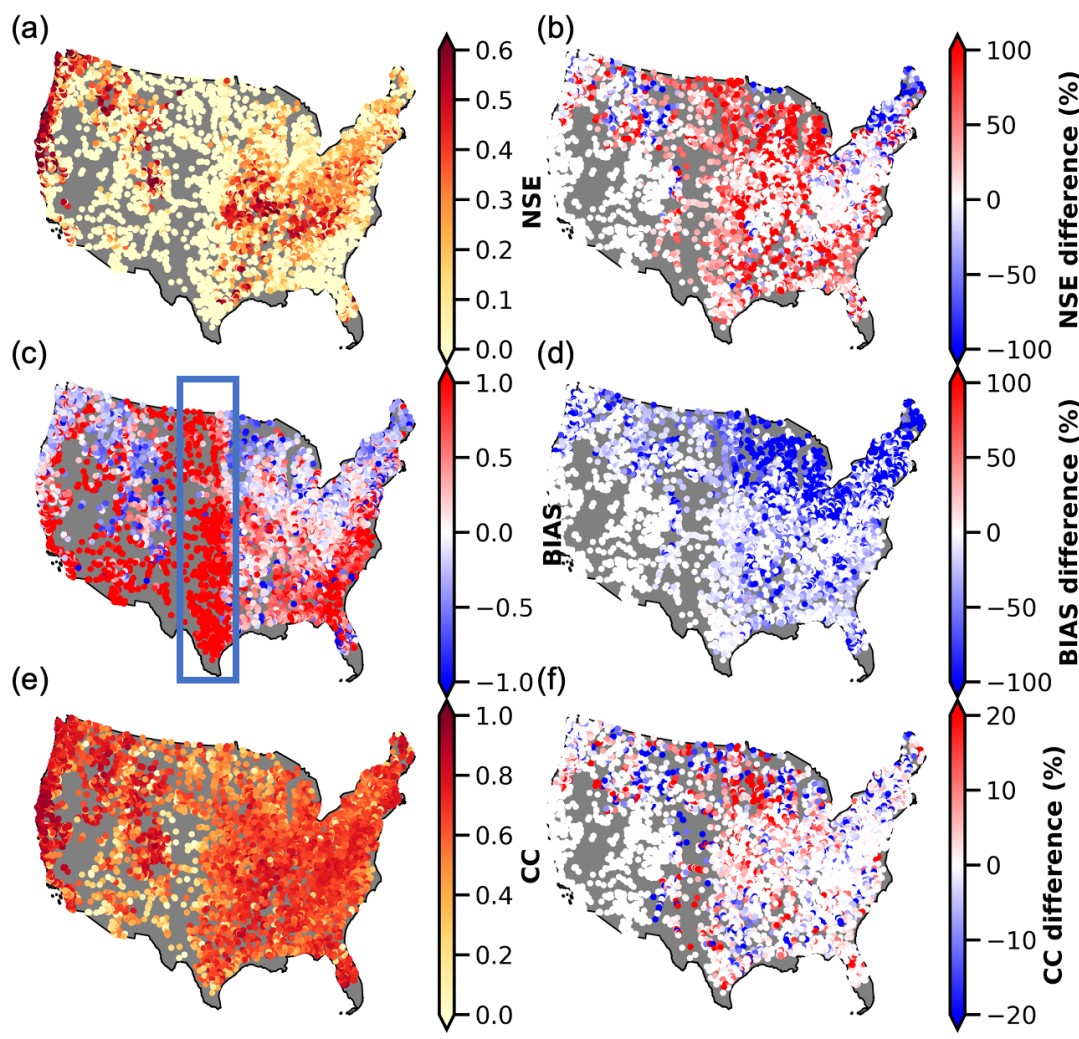

650

**Figure 6. Spatial map of model performance with the lake (left column) and the difference between with and without lake simulation (right column). (a): NSE scores; (b) NSE differences (results with lake minus results without lake); (c) BIAS; (d) BIAS difference; (c) Correlation Coefficient (CC); (f) CC difference. The blue box in (c) highlights the region where high positive BIAS is present.**



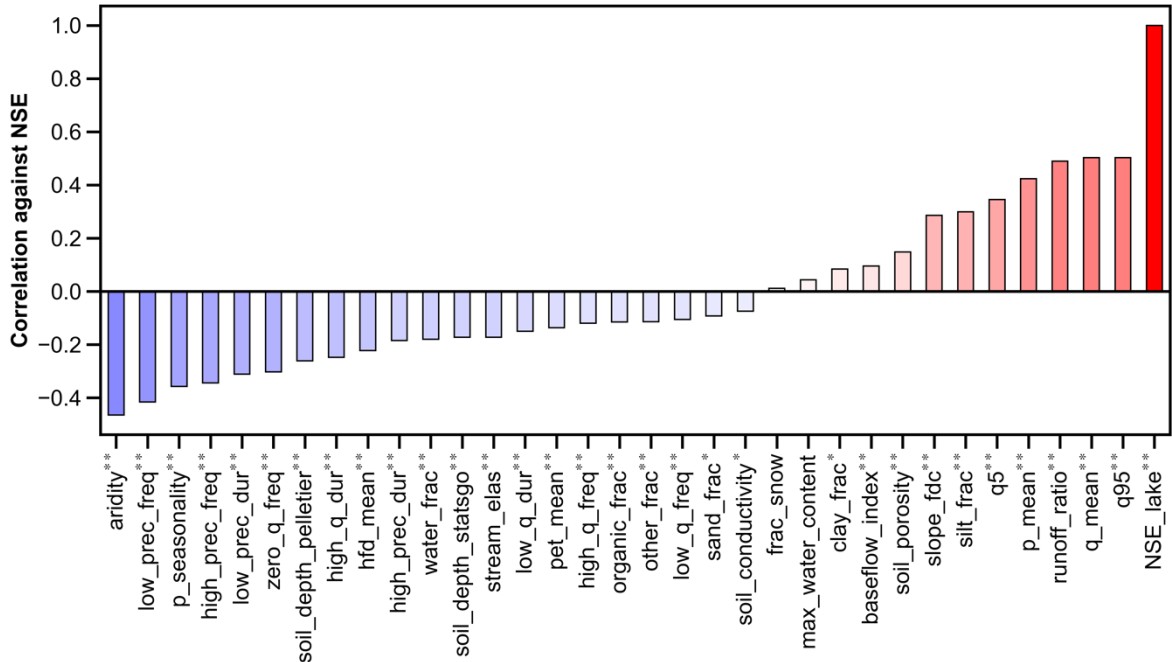

655

**Figure 7. Bar plot of Spearman Correlation between 34 environmental indices (including NSE itself) from the U.S. CAMELS dataset and NSE scores simulated by CREST-VEC. Asterisk (*) signifies the statistically significant correlation between represented index with NSE (p-value < 0.05); Two asterisks (**) signify very significant correlation (p-value <0.01).**



**Figure 8. Boxplot of catchment attributes from CAMELS dataset in poorly performed regions and overlaid with CAMELS catchments: (a) the Great Plains and (b) the Southeast. The gray shaded area represents the value range for each feature across all CAMELS gauges.**



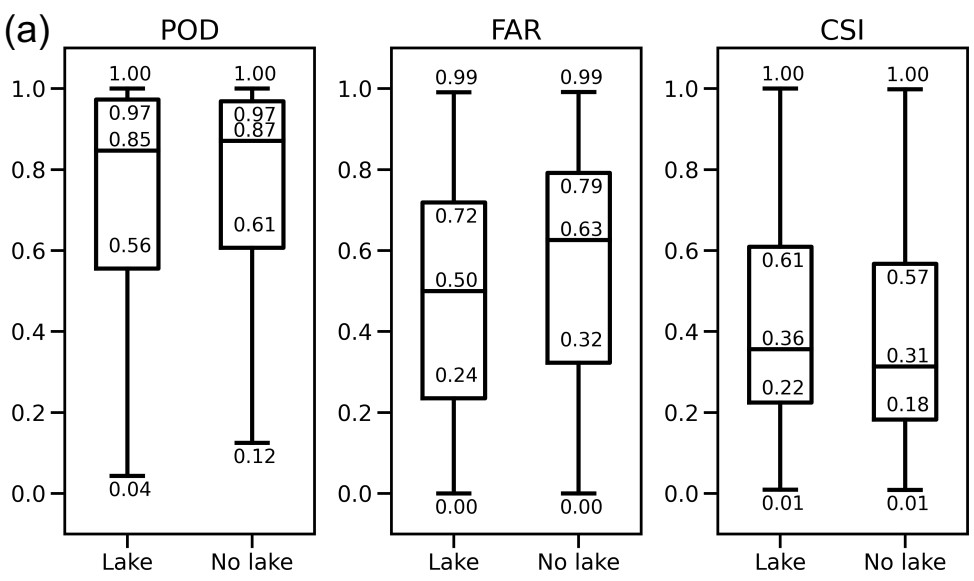

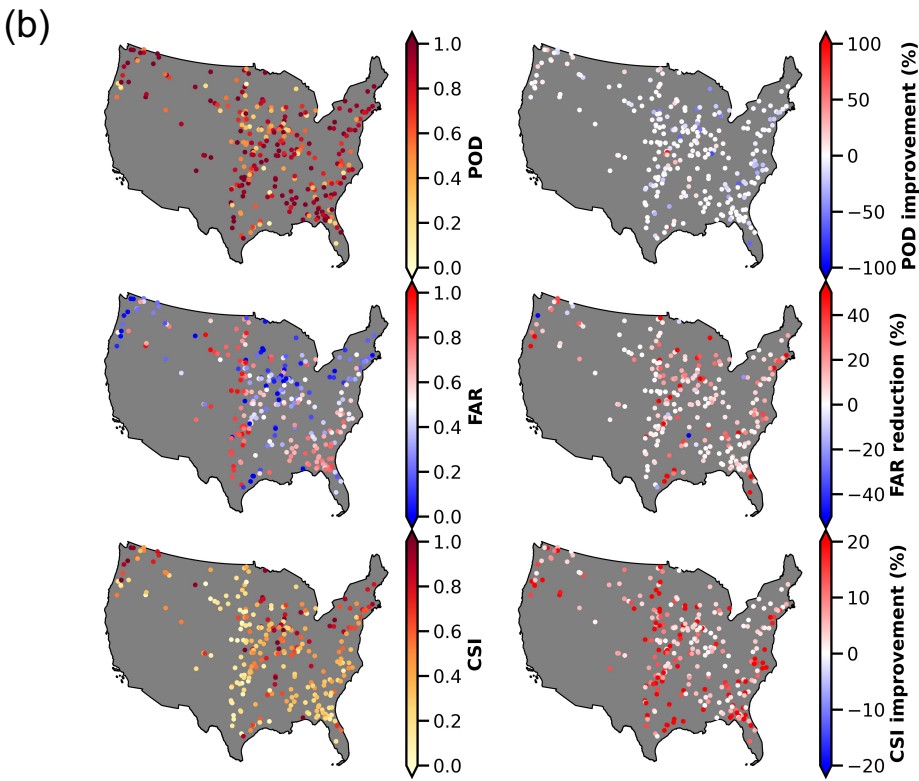

**Figure 9. Flood detection performance comparing lake and no lake simulation. (a) Similar to Figure 5, but for flood detectability; (b) Similar to Figure 6, but for flood detectability.**



Figure 10. Five case examples of streamflow time series at gauges downstream of lakes: (a) St. Johns River near Sanford, FL; (b) St. Johns River near Cocoa, FL; (c) St. Johns River near De Land, FL; (d) Big Muddy River at Plumfield, IL; (e) Mississippi River at Clinton, IA. Images courtesy of © Google Map.