# Peer review of "CREST-VEC: A framework towards more accurate and realistic flood simulation across scales"

_Geoscientific Model Development, 2022_

## Referee Comment (RC1)

**General Comments**

The authors provide a description of a new hydrologic model framework, CREST-VEC, which couples CREST and vector-based routing framework, mizuRoute, and sub-surface runoff and lake schemes are also incorporated. Compared with the raster-based routing scheme implemented in the original CREST, the new framework shows a significant improvement in computational efficiency and reproducibility in river discharge. Therefore, the proposed model has an essential potential for a flood forecast and alert system, as the authors suggested, and the manuscript is informative for the readers of this journal. However, since two contexts (speed-up of the model and introduction of new physical processes) are mixed in the manuscript, it requires some revision of the manuscript and experimental configurations for a well-structured article.

**Specific Comments**

For the abstract and corresponding sections:

The authors are highly requested to discuss 1) vectorization of the routing scheme and the improvement in computational efficiency (and the ensemble simulation, as they suggested) and 2) the introduction of new physical processes, the change in reproducibility of river discharge, and the false alert problem separately. Therefore, the article should be divided into two parts for the sake of readability. However, if they are still to be published as one paper, some parts need to be reorganized. For example, the introduction should describe the significance of the subsurface flow and lake routing for the flood forecast.

The authors used computation time per step as a measure of computational efficiency. However, the number of computation steps varies depending on time-step width and constraints such as CFL conditions in some models (not sure if this is the case with your model). Therefore, it would be more appropriate to use the time it takes to compute a given period on a given spatial resolution (e.g., one year) rather than the computation time per step. The number of parallels and parallel efficiency are also important indicators for comparing computation algorithms.

In my understanding, the routing scheme calculates the time lag from runoff from the land surface to downstream. So, why does the bias change between CREST and CREST-VEC, as shown in Figure 4 and Table 1? Or is this bias calculated for peak flows? (One possible factor is the use of externally derived reservoir storage. However, such modifications should not be applied even if they improve the model accuracy since it is difficult to discuss the impacts of the model update when the water budget in the overall model is changed.)

**1 Introduction**

Mainly for line 69: Even if a numerical model can be run at a fine spatial resolution in a realistic amount of time, it does not necessarily mean that the physics assumed in the model hold at the resolution. For example, a 1-D river model with a 100-m resolution is difficult to apply directly to a wide river such as the Amazon River, even if it can be run. The authors' work is technically excellent, but the validity of the model physics should be discussed in the discussion section.

Line 84-85: Raster-based and vector-based models are reviewed in detail in the introduction section. However, information relevant to the research question listed here should be added. If previous studies on the impact of subsurface processes and lake considerations on a river model are presented, the reader will better understand the motivation for the study.

**3.2 CONUS simulation**

Line 263-289: This discussion using the basin attributes suggests a significant uncertainty in the land surface process, not the routing schemes. Remove it, or explicitly describe the contribution of the CREST-VEC (+subq+lake) to the discussion compared with the results when the same analysis is applied to the original CREST.

**3.3 How likely are floods falsely detected?**

Line 315-317: Previous research also reported the incorporation of a lake scheme mitigates the seasonal variability in the river discharge (Tokuda et al., *GMD*, 2021).

**Technical Corrections**

**1 Introduction**

Line 25: Add "." after "1".

Line 38: "the most time-consuming". The expression is too strong and should be loosened. It is because atmospheric models and (even within the framework of a land surface model) land surface models that account for 2-D groundwater flow are computationally expensive.

Line 57: 2011; Yamazaki et al., 2011;) remove the last ";".

**2. Data and methods**

**2.1 Hydrography data**

Line 92: Remove "MERIT-Hydro" before Lin et al. (2019), which corresponds to Yamazaki et al. (2019).

Line 94: Add the reference for the MERIT DEM (Yamazaki et al., 2017).

**2.2 Forcing data**

Line 118: "rainfall" implies snowfall is excluded?

**2.5 CREST-VEC**

Line 174: The reservoir operation is incorporated in the mizuRoute but does not the CREST-VEC consider it?

**3. Results**

**3.1 Case study: Houston region**

Line 210: Related to the major comment, why does the CREST-VEC overestimate the peak compared to the original CREST?

Line 219 (Figure 3): Make the orange lines visible.

**3.3 How likely are floods falsely detected?**

Line 321: The contribution of the routing scheme to the flood alert system is an essential topic for our society. However, additional analysis is needed since the current analysis is highly affected by the bias of the runoff, and we can not detect the impacts of the new routing and lake schemes from the results.

**4. Discussion**

**4.1 Vector vs. Raster routing**

Line 353-355: Which result (or previous research?) suggests this?

Line 359-361: What is the relationship between the one-to-many river network and the following sentences? What does "on the other hand" mean?

90 Line 363-368: What is the difference in the sub-grid routing scheme between raster- and vector-based approaches? The section title is "vector vs. raster", but the sub-grid scheme represents the hydrodynamics within each grid, and it has nothing to do with the vector or raster approaches.

**4.2 Room for improving large-scale hydrologic simulation**

95 Line 370-396: The paragraphs are too long to discuss the uncertainty caused by the IRF scheme. Make them shorter. (Or, do you have any plan for the calibration with the latest, faster CREST-VEC model?)

**4.3 How to operate flood forecasting with regulated flow?**

Line 408 (The section title): Since the reservoir operation is not considered in this experiment, "with a lake scheme" is more
100 appropriate instead of "with regulated flow".

Line 411-412: Which process should be improved in the current lake scheme? The improvement plan should be discussed in the previous section (4.1).

105 Line 415-419: Does not this approach cause the "cry wold" effect?

---

## Author Comment (AC1)

General Comments

The authors provide a description of a new hydrologic model framework, CREST-VEC, which couples CREST and vectorbased routing framework, mizuRoute, and sub-surface runoff and lake schemes are also incorporated. Compared with the raster-based routing scheme implemented in the original CREST, the new framework shows a significant improvement in computational efficiency and reproducibility in river discharge. Therefore, the proposed model has an essential potential for a flood forecast and alert system, as the authors suggested, and the manuscript is informative for the readers of this journal. However, since two contexts (speed-up of the model and introduction of new physical processes) are mixed in the manuscript, it requires some revision of the manuscript and experimental configurations for a well-structured article.

Response:

Thank you for your comments and suggestions. We made changes to restructure our manuscript as suggested. Specific changes are listed below.

Specific Comments

For the abstract and corresponding sections:

The authors are highly requested to discuss 1) vectorization of the routing scheme and the improvement in computational efficiency (and the ensemble simulation, as they suggested) and 2) the introduction of new physical processes, the change in reproducibility of river discharge, and the false alert problem separately. Therefore, the article should be divided into two parts for the sake of readability. However, if they are still to be published as one paper, some parts need to be reorganized. For example, the introduction should describe the significance of the subsurface flow and lake routing for the flood forecast.

Response:

Thanks for your suggestions. We made following changes to our manuscript.

In the **Abstract**, we specifically mentioned the objective of this study:

"Large-scale (i.e., continental and global) hydrologic simulation is an appealing yet challenging topic for the hydrologic community. First and foremost, model efficiency and scalability (flexibility in resolution and discretization) have to be prioritized. Then, sufficient model accuracy and precision are required to provide useful information for water resources applications. Towards this goal, we craft two objectives to improve US current operational hydrological model: (1) vectorized routing and (2) improved hydrological processes. This study presents a hydrologic modeling framework CREST-VEC that combines a gridded water balance model and a newly developed vector-based routing scheme."

In the **Introduction**, we added one paragraph to introduce the importance of subsurface routing and lake routing specifically to flood prediction.

L.90-97: "To date, modern vector-based routing models such as RAPID and mizuRoute have neglected the subsurface routing, which is either assumed to be minimum (Mizukami et al., 2020) or treated in the way same as surface routing (Lin et al., 2019; Yang et al., 2021). However, subsurface routing is an important hydrologic process and dominates over regions that feature intermittent flow

behaviors (Freeze, 1972). For flood simulation, ignoring subsurface routing could underestimate the peak flow and miscalculate the flood timing, both of which directly affect decision-making processes. An equally important research thrust is the representation of lakes and reservoirs in vector formats, since they markedly alter flow response not only at local scale, but also downstream rivers. One significant function of lakes and reservoirs in the US is for flood control, so simulation without incorporating such process is likely to result in falsely issued flood warnings."

The authors used computation time per step as a measure of computational efficiency. However, the number of computation steps varies depending on time-step width and constraints such as CFL conditions in some models (not sure if this is the case with your model). Therefore, it would be more appropriate to use the time it takes to compute a given period on a given spatial resolution (e.g., one year) rather than the computation time per step. The number of parallels and parallel efficiency are also important indicators for comparing computation algorithms

Response:

Thanks for your suggestions. In this study, we used average time taken per step for the whole simulation period – 5 years. Since the legacy model CREST does not have parallel capacity, we only compared single-core performance in this study. Parallel efficiency has been heavily discussed in Mizukami et al. (2021). In addition to comparing routing efficiency, we also take into account the total computational cost listed in Table 1, from which the new scheme is about six times faster than the operational model, even at higher resolution.

Table 1. Statistical comparison of model performance over the continental U.S. Bolded numbers indicate the best metrics among the three model configurations. The computational speed is calculated as an average speed over a whole simulation period.

| Metrics | Gridded CREST (Flamig et al., 2020) | CREST-VEC (w/o lake) | CREST-VEC (w/ lake) |
|---|---|---|---|
| Simulation resolution | 1 km | **90 m** | **90 m** |
| Total computational cost (hours) | 149.2 | **29.9** | 32.96 |
| Computational Speed for routing (sec/step) | 7.2 | **0.35** | 0.37 |
| Max NSE | 0.71 | **0.87** | **0.87** |
| Median NSE | -0.06 | 0.12 | **0.18** |
| % gauges NSE>0 | 41.8 % | 50.6 % | **56.2 %** |
| Max CC | **1.0** | 0.96 | 0.96 |
| Median CC | 0.40 | **0.67** | **0.67** |
| Median bias | **9%** | 27% | 17% |

In my understanding, the routing scheme calculates the time lag from runoff from the land surface to downstream. So, why does the bias change between CREST and CREST-VEC, as shown in Figure 4 and Table 1? Or is this bias calculated for peak flows? (One possible factor is the use of externally derived reservoir storage. However, such modifications should not be applied even if they improve the model accuracy since it is difficult to discuss the impacts of the model update when the water budget in the overall model is changed.)

Response:

Thanks for your comments. The bias differences of streamflow shown between CREST and CREST-VEC originate from three parts. First, CREST and CREST-VEC use different routing scheme. CREST adopts a simplified kinematic wave routing (Vergara et al., 2016), while CREST-VEC uses a Unit Hydrograph routing (Mizukami et al., 2016). These two routing schemes in principle have different physical representations. Second, in Figure 4 and Table 1, we listed results both with and without lake module, and certainly results with lake module have less bias with considerable water being held in lakes. Third, Specifically for Figure 4, the CREST model is heavily calibrated utilizing the DREAM optimizer. So, the peak flow is calibrated against observed streamflow. However, the results with CREST-VEC are not calibrated because of large-scale simulation, which is considered as a future work. We included more calibration information in our main text:

L.233-236: "For the CREST model with gridded routing, we calibrate the model using the DREAM (Differentiable Evolution Adaptive Metropolis) optimization (Vrugt et a., 2009) from 2016-06-01 to 2017-06-01 at an hourly time step and perform evaluation from 2017-06-01 to 2020-01-01. The NSCE is used as objective function for calibration, and model is warmed up for one year from 2016-06-01 to 2017-06-01."

**1 Introduction**

Mainly for line 69: Even if a numerical model can be run at a fine spatial resolution in a realistic amount of time, it does not necessarily mean that the physics assumed in the model hold at the resolution. For example, a 1-D river model with a 100-m resolution is difficult to apply directly to a wide river such as the Amazon River, even if it can be run. The authors' work is technically excellent, but the validity of the model physics should be discussed in the discussion section.

Response:

Thanks for your comments. We admit current limitations in model physics in the discussion section and pasted here.

L. 699-706: "The CREST-VEC model, by no means, represent all physical hydrological processes. Instead, it is a conceptual flood forecast model that aims to deliver timely flood information to stakeholders, decision-makers, and broader users. We admit that some processes such as vadose zone modelling, snow melting, hillslope routing, in-channel river routing, and reservoir operations are simplified, and some processes such as vegetation and groundwater modelling are missing for the current version. Since there is always the trade-off between model complexity and efficiency, we hope to continuously push the envelope of this front to optimize real-time flood forecast system."

Line 84-85: Raster-based and vector-based models are reviewed in detail in the introduction section. However, information relevant to the research question listed here should be added. If previous studies on the impact of subsurface processes and lake considerations on a river model are presented, the reader will better understand the motivation for the study.

Response:

Thanks for your suggestions. We tailored four research questions to be reflective of the overall information we want to convey. Because previously the model performance has not been evaluated

with and without subsurface routing and lake routing, especially regarding the linkage to flood warnings, we emphasize the importance of the two in our research questions.

L. 103-105: "Four questions are posed in this regional case study: (1) What is the performance gains for CREST-VEC compared to CREST model? (2) Does the included subsurface routing improve model performance? (3) Can a simple natural lake simulation improve model performance in a downstream urban area? and (4) How does CREST-VEC model adopt to flood warnings?"

**3.2 CONUS simulation**

Line 263-289: This discussion using the basin attributes suggests a significant uncertainty in the land surface process, not the routing schemes. Remove it, or explicitly describe the contribution of the CREST-VEC (+subq+lake) to the discussion compared with the results when the same analysis is applied to the original CREST.

Response:

Thanks for your suggestions. As it is also brought up by another reviewer, we deleted this section for brevity.

**3.3 How likely are floods falsely detected?**

Line 315-317: Previous research also reported the incorporation of a lake scheme mitigates the seasonal variability in the river discharge (Tokuda et al., GMD, 2021).

Response:

Thanks for your suggestions. The reference has been added.

L. 565-566: "Additionally, previous research reported that simulation results with lake module mitigate the seasonal variability of the river discharge (Tokuda et al., 2021)."

**Technical Corrections**

Line 25: Add "." after "1"

Response: Corrected.

Line 38: "the most time-consuming". The expression is too strong and should be loosened. It is because atmospheric models and (even within the framework of a land surface model) land surface models that account for 2-D groundwater flow are computationally expensive.

Response:

Thanks for your suggestions. We modified this sentence to:

L. 41-42:"It is an inseparable component in hydrologic simulation to redistribute and exchange water between compartments and is also relatively time-consuming"

Line 57: 2011; Yamazaki et al., 2011;) remove the last ";".

Response: Corrected.

Line 92: Remove "MERIT-Hydro" before Lin et al. (2019), which corresponds to Yamazaki et al. (2019)

Response: Corrected.

Line 94: Add the reference for the MERIT DEM (Yamazaki et al., 2017).

Response: Thanks for your suggestions. This reference is added.

Line 118: "rainfall" implies snowfall is excluded?

Response: Thanks for your comments. All types of precipitation are included. We changed "rainfall" to "precipitation" throughout the text.

Line 174: The reservoir operation is incorporated in the mizuRoute but does not the CREST-VEC consider it?

Response: It is included as an option in CREST-VEC, but we did not turn it on in this study for performance analysis because it complicates model configuration.

Line 210: Related to the major comment, why does the CREST-VEC overestimate the peak compared to the original CREST?

Response: Thanks for your comments. We have mentioned three reasons as mentioned in your major comment.

Line 219 (Figure 3): Make the orange lines visible.

Response: Thanks for your suggestions. We replotted this Figure to make it visible.

[Figure]

Line 321: The contribution of the routing scheme to the flood alert system is an essential topic for our society. However, additional analysis is needed since the current analysis is highly affected by the bias of the runoff, and we can not detect the impacts of the new routing and lake schemes from the results.

Response:

Thanks for your comments. We agree that high bias exists in current model settings and such errors can mislead our flood alert system. This work focuses more on the coupled framework which has a potential to replace current operational flood forecast framework. We acknowledge that there are more efforts to be done in the future to correct such bias from atmospheric forcing or model physics. As mentioned previously, we have mentioned our model physics is not perfect – some processes are simplified/missing. Beyond model physics, we need large-scale model calibration efforts to account for streamflow adjustment, as outlined in Discussion Section. But opportunistically, we see an overall improvement in flood prediction as compared to original model.

Line 353-355: Which result (or previous research?) suggests this?

Response:

Thanks for your comments. Our previous sentence is a bit overstating. We have now modified this sentence to:

"In terms of results against observations, the CONUS-wide performance is improved regarding NSE values"

Line 359-361: What is the relationship between the one-to-many river network and the following sentences? What does "on the other hand" mean?

Response:

Thanks for your comments. We intend to show both sides of vector and raster-based routing. On one hand, vector-based routing has xxx deficiencies. One the other hand, raster-based routing has xxx deficiencies. Since this "on the other hand" is confusing, we have deleted it from our main text.

Line 363-368: What is the difference in the sub-grid routing scheme between raster- and vector-based approaches? The section title is "vector vs. raster", but the sub-grid scheme represents the hydrodynamics within each grid, and it has nothing to do with the vector or raster approaches.

Response:

Thanks for your comments. We have deleted this sentence for clarity.

Line 370-396: The paragraphs are too long to discuss the uncertainty caused by the IRF scheme. Make them shorter. (Or, do you have any plan for the calibration with the latest, faster CREST-VEC model?)

Response:

We have shortened this paragraph to only tie to our model results. For future calibration, a feasible approach is through parameter regionalization. It requires careful calibration on a set of unique catchments (by catchment signatures) such as CAMELS dataset. Then, we can transfer calibrated parameters to other regions by feature proximity. Similar work can be found in Mizukami et al. (2017).

L.719-744: "Hydrologic calibration is powerful to boost model accuracy, yet large-scale models oftentimes suffer from the complexity that impedes credible model calibration. River routing schemes and their parameters can affect streamflow simulations especially at fine time scale such as sub-daily (Mizukami et al., 2021). Our current study used IRF scheme in which impulse response function is derived from diffusive wave equation (see Lohman et al., 1996; Mizukami et al., 2016) and includes two parameters – diffusivity and celerity. These parameters need to be included in calibration in addition to the hydrologic model parameters. Furthermore, to fully understand routing model impact on streamflow simulations, it is necessary to consider other routing schemes including diffusive wave as well as kinematic wave."

Line 408 (The section title): Since the reservoir operation is not considered in this experiment, "with a lake scheme" is more appropriate instead of "with regulated flow"

Response:

Thanks for your suggestions. We agree that "with a lake scheme" is more appropriate in this context. We have changed this title to "Towards improved flood forecasting with lake routing"

Line 411-412: Which process should be improved in the current lake scheme? The improvement plan should be discussed in the previous section (4.1).

Response:

The representation of lake routing should and can be improved because it is now simple. A sophisticated lake routing module including more human control should be updated. We have discussed this in Section 4.1.

L. 704-705: "For the lake module, we expect to include more sophisticated multi-layer decision processes instead of a level-pool process. Lake evaporation is another important factor to be considered for improved water balance."

Line 415-419: Does not this approach cause the "cry wolf" effect?

Response:

This approach still causes "cry wolf" effect, but at a less rate than original flood forecast framework. A large contribution factor is the lake module which by its purpose is to control floods. We see that including lake routing can significantly reduce such "cry wolf" effect.

**A list of references mentioned in this response letter:**

Mizukami, N., Clark, M. P., Gharari, S., Kluzek, E., Pan, M., Lin, P., Beck, H. E., Yamazaki, D.: A vector-based river routing model for Earth System Models: Parallelization and global applications. J. Adv. Model. Earth Syst., 13, e2020MS002434, https://doi.org/10.1029/2020MS002434, 2021.

Vergara, H., Kirstetter, P., Gourley, J.J., Flamig, Z.L., Hong, Y., Arthur, A., and Kolar, R: Estimating a-priori kinematic wave model parameters based on regionalization for flash flood forecasting in the Conterminous United States, 541, 421-433, doi: 10.1016/j.jhydrol.2016.06.011, 2016.

Mizukami, N., Clark, M. P., Sampson, K., Nijssen, B., Mao, Y., McMillan, H., Viger, R. J., Markstrom, S. L., Hay, L. E., Woods, R., Arnold, J. R., and Brekke, L. D.: mizuRoute version 1: a river network routing tool for a continental domain water resources applications, Geosci. Model Dev., 9, 2223–2238, https://doi.org/10.5194/gmd-9-2223-2016, 2016.

Mizukami, N., Clark, M. P., Newman, A. J., Wood, A. W., Gutmann, E. D., Nijssen, B., Rakovec, O., and Samaniego, L.: Towards seamless large-domain parameter estimation for hydrologic models, Water Resour. Res., 53, 8020– 8040, https://doi.org/10.1002/2017WR020401, 2017.

Yamazaki, D., Ikeshima, D., Tawatari, R., Yamaguchi, T., O'Loughlin, F., Neal, J. C., Sampson, C. C., Kanae, S., and Bates, P. D.: A high-accuracy map of global terrain elevations, Geophys. Res. Lett., 44, 5844–5853, https://doi.org/10.1002/2017GL072874, 2017.

---

## Author Comment (AC2)

This manuscript introduces a vector-based routing scheme (mizuroute) into the CREST hydrologic model. Additionally, the authors augment the routing scheme by adding a new subsurface routing scheme and a lake module. The new model is then tested for multiple scenarios. The manuscript is well-written and the conclusions drawn are consistent with the presented results. I have the following suggestions:

Response:
Thanks for your suggestions and comments. We have modified our manuscript as detailed below.

As it stands, the manuscript does not do a good job of separating the two distinct contributions - subsurface routing and lake module - in the text. The figures show clear delineation. I would suggest adding separate sections with detailed descriptions for both the subsurface routing scheme and the lake module.

Response:
Thanks for your suggestions. We have extended paragraphs to introduce the newly added lake module and subsurface routing.

In Section 2.4, we added following description of how to model subsurface flow.

L. 191-199: "In this study, we enable an option to turn on or off subsurface routing as defined in the model configuration file. Similar to surface runoff routing, the subsurface flow is routed using the IRF scheme but with much slower velocity and reduced magnitude. We use a two-parameter Gamma distribution function to materialize the IRF method as shown in eq. 1.

$$y(t) = \frac{1}{\Gamma(a)\theta^a} t^{a-1} e^{-\frac{t}{\theta}}$$

Where t is the time variable, a is a shape parameter, and $\theta$ is a time-scale parameter. Both a and $\theta$ determine the flood peaking time and flashiness. After calculating instantaneous rates based on gamma function, we use a convolution to compute flow rates Q at time t. R(t-s) is the (sub)surface runoff at time (t-s), and s is an increment of time from 0 to tmax (also denoted as the time window). The default values of a and $\theta$ for hillslope surface routing are set to 2.5 and 8000. For subsurface flow routing, the a and $\theta$ are 10 and 86400, respectively.

$$Q(t) = \int_0^{tmax} y(t) \times R(t-s)ds"$$

In Section 2.5, we have enriched descriptions of lake module:

L. 215-225: "We use the IRF scheme in this study for both terrain routing and channel routing in this study and activate the lake model with the Döll et al. (2003) lake model. The parameters for lake parameters such as the outflow coefficient $a$ and exponent $b$ of eq.3, are based on suggested values in Döll et al. (2003) and Gharari et al. (2022). For lakes that have monitored storage provided by the US Geological Survey (USGS), we directly insert storage time series into the model. As reservoir operation is not considered in this study, we exclude observed streamflow that is regulated by reservoirs and regulated lakes, as shown in Fig. 1c. So, only results from natural lakes, which account for 98% of US lakes or reservoirs, are considered valid for

statistical comparison. To initialize model states, especially for initial lake volumes, we warm up the CREST-VEC model from 1948 to 2014 using the GLDAS forcing (Global Land Data Assimilation System) at a daily time step.

$$Q_{out} = a \times S_f \times (S_f/S_{f,max})^b,$$

where $a$ and $b$ are the outflow coefficient (1/day) and exponent, respectively; $S_f$ is the actual lake storage (m$^3$); $S_{f,max}$ is the maximum lake storage (m$^3$)."

The results section is well structured and clearly flows from regional application to a continental use case, and finally a flood forecasting example.
However, the discussion section is insufficient. Section 4.2 is largely unnecessary as the paper does not deal with hydrologic simulation at all, unless the authors want to test ensemble simulation with varying catchment processes.

Response:
Thanks for your suggestions. In a nutshell, we still consider this framework CREST-VEC as hydrologic simulation, since it has both water balance module and routing module. This section intends to lay out future strategies to improve large scale hydrologic simulation after we see commonly poor scores in hydrologic models in regions like the Great Plains. The ensemble simulation could make a difference but has been so far being ignored in large-scale hydrologic simulation. However, we agree that previous version overstated some sentences that are out of context, so we have shortened this section and only kept parts that are relevant to our study.

Additionally, more discussion of the results from the flood forecasting example is needed. The authors need to contextualize the results within the large body of flood forecasting literature. In addition, the reasons for the improved FAR and/or reduced POD is not adequately addressed. I would suggest providing concrete mechanistic reasons for both improved FAR and reduced POD.

Response:
Thanks for your suggestions. After integrating other three reviewers' comments, we realize that suggesting flood forecasting is off our intention and distract the main focus of this work. We chose not to expand on this topic. However, we agree on your suggestion that more mechanistic reasons for improved FAR and reduced POD are needed. We have made such revisions to our main text.

L.566-573: "The decrease in FAR values implies five instances: (1) decrease in false alarms while hits remain the same; (2) increase in hits while false alarms remain the same; (3) decrease in false alarms while increase in hits; (4) decrease in both false alarms and hits; (5) increase in both false alarms and hits. We find that however POD values decrease from 0.87 (without lake) to 0.85 (with lake), which indicates that both hits and false alarms are decreasing, but false alarms decrease at a higher rate. This is due to the reduced, simulated flood peak, and consequently less hits in flood forecasts but meanwhile less falsely alarmed floods. As most studies focus on flood detection, they inevitably arrive at more falsely detected floods. Too many false alarms could make people disregard the warnings, despite a real threat, causing the "cry wolf" effect."

---

## Author Comment (AC3)

I have finished my review of the paper "CREST-VEC: A framework towards more accurate and realistic flood simulation across scales", by Li et al., submitted to Geoscientific Model Development. This paper outlines the integration of two existing models – the gridded hydrological model CREST and the vector-based routing tool mizuRoute. The authors examine the (significant) improvement in computational cost associated with using a vector-based routing tool and explore the impact of including or not including lakes on simulations of 5350 streamflow hydrographs across the U.S., with impacts evaluated using standard model fit metrics (e.g., NSE/bias) and using flood detection measures (POD, FAR, and CSI). Lastly, the author examine the correlation of absolute model performance against environmental indices from the CAMELS database.

In general, I found this paper to be a bit disorganized and missing a number of important details. It uses less-than-ideal approaches for comparing populations of model runs and does not seem (to me) to represent a significant contribution either as a model development or in presenting new insights into models more generally. I document below a number of major issues with the paper which should be addressed prior to publication.

Response:

Thank you for your constructive comments and suggestions. Our responses to your comments are listed below.

- The title of this paper implies it may be a discussion of a fundamentally new model or modelling framework. However, it is the merger of an existing routing model which has already been coupled to multiple existing hydrological models (including gridded models) with yet another hydrological modelling code. The details of this integration are generally missing, so it is difficult to evaluate the challenge in performing this integration, whether it is file-based coupling or a single compiled code, whether anything other than the spatial aggregation of runoff is causing the computational improvements in the shift from CREST to CREST-VEC. It does not appear that this model development effort in and of itself is a contribution. If it is, then the authors need to demonstrate why.

Response:

Thank you for your comments. This framework is in fact not only a merger of existing codes, but we did improvements to the codes and added critical subsurface routing. Prior to this point, the evaluation of both subsurface routing and lake operation have not been comprehensively evaluated over the US. The scientific scope of this study is to draw attention to falsely alarmed floods if these two components in hydrologic framework are missing. We expanded the model integration details and described more about the newly added components.

L.208-214: "The framework has loosely coupled by two models written in different programming languages. A bash file calls three executables after model compilation subsequently (CREST-EASYMORE-mizuRoute). The input files for this model chain include forcing data (gridded precipitation, potential evaporation, and temperature), topography data (gridded digital elevation model, flow direction, flow accumulation, river network topology, and hydrologic response unit), and configuration files. The topography data can be accessed from the HydroSHEDS website which consists of grid-based and vector-based topography data."

More details shown in subsurface routing is described.

L. 191-199: "In this study, we enable an option to turn on or off subsurface routing as defined in the model configuration file. Similar to surface runoff routing, the subsurface flow is routed using the IRF scheme but with much slower velocity and reduced magnitude. We use a two-parameter Gamma distribution function to materialize the IRF method as shown in eq. 1.

$$y(t) = \frac{1}{\Gamma(a)\theta^a} t^{a-1} e^{-\frac{t}{\theta}}$$

Where t is the time variable, a is a shape parameter, and θ is a time-scale parameter. Both a and θ determine the flood peaking time and flashiness. After calculating instantaneous rates based on gamma function, we use a convolution to compute flow rates Q at time t. R(t-s) is the (sub)surface runoff at time (t-s), and s is an increment of time from 0 to tmax (also denoted as the time window). The default values of a and θ for hillslope surface routing are set to 2.5 and 8000. For subsurface flow routing, the a and θ are 10 and 86400, respectively.

$$Q(t) = \int_0^{tmax} y(t) \times R(t - s)ds"$$

- The authors use results from thousands of model runs in order to assess the benefit of including lakes in vector-based routing. The effort to do this is not trivial, and I applaud the authors for their ambition in the size of this computational experiment. However, there are some issues with both the magnitude of the contribution and the means of assessing the results. About the question of whether simulating lakes is useful, the inclusion of lakes has already been demonstrated to improve vector-based routing results (Han et al., 2020), though perhaps not in as rigorous of a manner as could potentially be done with this massive model set. However, there are several fundamental issues with the analysis herein. Firstly, the model in general (before and after lakes are included) performs very poorly, with median NSEs on the order of 0.3, and 45% of simulations having NSEs below zero. The authors make no attempt to reduce their analysis to focus on models with that satisfy some adequate base performance. They also spend most of their time discussing improvements to the median, which means that much of the performance can easily be due to changing very poorly performing models to slightly less terrible models. They miss the opportunity to statistically compare the probability distributions of metrics to see whether the distributions are functionally different. At the very least, I would ask that they compare median change in NSE (a more adequate metric for evaluating improvements) rather than the change in median NSE.

Response:

Thanks for your comments.

**First**, regarding the model performance, we compared our results with other studies since it is hard to perceive the model performance without relative comparison. One recent study by Tijerina et al., (2021) did continental simulation using two state-of-the-science hydrologic models - ParFlow and WRF-Hydro, and they used two indicators to describe (CC and Bias) as shown in Figure 1. Parflow has 56% gauges identified as "good" (low bias and good shape; green color), and the National Water Model (NWM) has 65%. For our results, 63.3% gauges have good agreements, in contrast to 61.2% of them without lake simulation. It is understandable since the NWM has been regularly calibrating their results as a part of their missions. We have lower than 10% gauges with low bias and poor shape

(purple color). Overall, our model results have comparable results with modern continental-scale hydrologic models. As one objective of this study, we want to examine the potential improvement from with-lake configuration on streamflow simulation over a wide range of hydrometrical and geographical settings in the CONUS, rather than provide some optimal model setup and parameterization at the CONUS scale, which we believe is way beyond our scope and several steps forward from the current CREST-VEC or any existing CONUS models (as shown here). As far as what qualifies 'an adequate base simulation', there may be some room for debate but should be some bottom-line principles: first, one should be clearly aware of the sources of uncertainties including forcing, model structure, parameterization, streamflow observation as the reference, etc. Optimization, though effective in improving the model performance, compensates uncertainties from the other sources simply via adjusting model parameters. This has been acceptable for operational purposes but is not appropriate for this study where a modification of model structure is introduced. Instead, we use an a-priori parameter set that was developed based on remote sensing datasets and also evaluated at the CONUS scale (Vergara et al., 2016). The physical base of these a-priori parameters set a solid foundation for examining the new with-lake configuration, thus should not be compromised via parameter tunning,

Tijerina, D., Condon, L., FitzGerald, K., Dugger, A., O'Neill, M. M., Sampson, K., et al. (2021). Continental hydrologic intercomparison project, phase 1: A large-scale hydrologic model comparison over the continental United States. Water Resources Research, 57, e2020WR028931. https://doi.org/10.1029/2020WR028931

Vergara, H., Kirstetter, P., Gourley, J.J., Flamig, Z.L., Hong, Y., Arthur, A., and Kolar, R: Estimating a-priori kinematic wave model parameters based on regionalization for flash flood forecasting in the Conterminous United States, 541, 421-433. https://doi.org/10.1016/j.jhydrol.2016.06.011, 2016.

[Figure]

Figure 1. Simulation results by Tijerina et al. (2021). Copyright (2021) American Geophysical Union.

[Figure]

Figure 2. Similar to Figure 1, but produced with CREST-VEC.

**Second**, we discussed also the poor-performing regions, especially over the Great Plains where the model bias is large. We mentioned the challenges to hydrologic modelling community since it is ubiquitous to continental simulations.

L. 330-342: "CREST-VEC with lake module in regions like the West Coast and Upper Mississippi River Basin have relatively good performance (NSE>0.4), yet over the Great Plains and East Coast, the model bias is high (BIAS>1), yielding low NSE scores. Similar issues are found in the literature with other models (Clark et al., 2008; Konben et al., 2020; Lin et al., 2019; Mizukami et al., 2017; Newman et al., 2015; Salas et al., 2017; Knoben et al., 2020; Yang et al., 2021; Tijerina et al., 2021). Taking the Great Plains as an example (highlighted box in Fig. 6c), the model physics of CREST-VEC does not correctly represent the real hydrologic processes by two means. First, the surface runoff (before routing) simulated by CREST-VEC is biased. We compare the annual surface runoff by CREST-VEC to the public community dataset GRFR (Global Reach-level Flood Reanalysis) in Fig. S1. The runoff in GRFR is simulated by the VIC model and undergoes stringent bias correction against observations via the discrete quantile mapping technique (Yang et al., 2021; Lin et al., 2019). There is a 116.3% higher surface runoff by the CREST-VEC than theirs, partly explaining the high BIAS and low NSE scores in such region. We suspect the singular bulk soil layer represented in the CREST model yields such systematic differences. Second, the missing representation of playas, small and rain-fed lakes that are prominent in the Great Plains, leads to falsely produced runoff"

Besides, we attributed model performance to basin attributes in the Southeast and found the CREST-VEC model performs poorly for intermittent rivers and deep soil depth, both of which are related to hydrologic model physics.

"Figure 8b shows an example of poor-performing gauges in the Southeast. Analogous to the Great Plains, the soil depths in the Southeast are considerably high (1.5 meters), leaving CREST model simulations problematic. Evapotranspiration (ET) in the Southeast is also one of the highest among the US climate divisions due to abundant precipitation, permeable soils, dense vegetation, and substantial soil radiation. Because the CREST model does not account for transpiration from vegetation nor solve the energy balance explicitly, the simulated evaporation rates may be lower than actual evaporation rates, resulting in higher effective rainfall and thus positive bias of streamflow.

Therefore, the missing hydrologic processes such as transpiration and infiltration-excess process in the Southeast are likely the causations of lower NSE scores."

**Third**, we not only evaluated the relative change in NSE, but also the distribution as the boxplots shown in Figs. 5 and 9. There are distribution shifts in the lower tails. As suggested, we modified the change in median metrics to median change in metrics throughout the main text.

- Most of this paper is thematically consistent – examining the benefits of using a vector-based routing model with lakes across the continental U.S.. The second half of section 3.2, however, evaluates the raw performance of the model (with lakes) against CAMELS environmental variables. This would make sense if the authors evaluated the benefit of including lakes (i.e., shift in NSE) against these variables, but as is, this analysis is really an examination of the quality of CREST's runoff estimates. I suggest that if this is to be retained, this section be re-cast to answer questions such as "under what environmental conditions is the inclusion of lakes more likely to be beneficial", which can be done by (e.g.) performing the analysis of figure 7 with improvements to NSE rather than the raw score.

Response:

Thank you for your suggestions. As brought up by another reviewer, we think it is out of context for the scope of this study, so we deleted this section for brevity.

- The calibration in the Houston case study is missing important details. How many parameters were calibrated? Which ones? Using what objective function? It also uses an unconventionally small 1 year calibration period and 2.5 year validation/evaluation period, with no reference to a run-up period.

Response:

Thanks for your comments. We added more descriptions into our main text. For the split-sample selection, there is no standard strategy. The intention here is to include Hurricane Harvey (2017-08 to 2017-09) in the validation period. Obviously, different split-sample strategy can cause uncertainties in model performance. The model warm-up period is from 2015-06-01 to 2016-06-01.

L.235-236: "The NSCE is used as objective function for calibration, and model is warmed up for one year from 2016-06-01 to 2017-06-01"

- How were the model parameters determined for the CONUS application?

Response:

The CONUS parameters of CREST are based off previous publications in which parameter regionalization (random forest) is used after calibrating hundreds of basins in the US. We added more information in the main text.

L.216-217:"The parameters for IRF routing are based on default values provided by Mizukami et al. (2016), and the lake parameters, such as the outflow coefficient a and exponent b of eq.1, are based on suggested values in Döll et al. (2003) and Gharari et al. (2022)."

L.308-309: "For this case, the CREST-VEC model parameters are based on the pre-configured CONUS-wide parameters, the same as the ones used in Flamig et al. (2020)."

- The justification for selecting model populations is missing – why/how were the 283 models in section 3.3 selected? Are these all of the gauges downstream of natural lakes in the dataset? What distance threshold defines 'downstream'? How were the 5 local cases discussed in figure 10 chosen – given the minimum NSE difference is > 0.55, these are not random samples, but rather seem to be cherry-picked to illustrate the most successful inclusion of lakes. It would be useful to see at least one model where inclusion of lakes degraded model performance, with speculation as to what circumstances might cause this.

Response:

Thanks for your suggestions. Those filtered gauges are downstream of natural lakes, which are filtered by finding gauges located at reach that is downstream lakes. The distance is varying, depending on the length of reach. The five gauges are randomly picked as they are all downstream of large reservoirs. For your suggestions, we replaced Fig.3e with a case that inclusion of lakes downgrade the model performance (NSCE: 0.3 for results with lake and 0.4 for results without lake). We see that despite the baseflow simulated by model w/ lake is closer to recorded values, the flood peaks are underestimated. On the contrary, model w/o lake better captures the flood peaks. We described this in our main manuscript.

L.590-592: "Figure 9e shows that although the model with lake produces better baseflow, it underestimates flood peaks, resulting in lower NSE values (0.3) than results without lake (0.4). It implies that parameters governing the lake outflow need to be improved."

[Figure]

Figure 3. Five case examples of streamflow time series at gauges downstream of lakes: (a) St. Johns River near Sanford, FL; (b) St. Johns River near Cocoa, FL; (c) St. Johns River near De Land, FL; (d)

Big Muddy River at Plumfield, IL; (e) Skagit River Near Concrete, WA. Images courtesy of Google Map.

- The false detection analysis of section 3.3 also suffers from the inclusion of all models, regardless of quality. Why even analyse the false detection performance with an NSE of 0.1? These models are already not fit for the task, so including them in the analysis could very well skew the results such that they imply performance improvements with lake inclusion, even if this performance improvement is meaningless (a NSE of -0.35 is not functionally better than an NSE of -0.55).

Response:

Thanks for your comments. For Section 3.3, we selected gauges only downstream of lakes, where we expect to see improved performance by including lake routing. First, the median NSE (CC/Bias) for lake included results become 0.3 (0.7/0.1) and 0.1 (0.6/0.3) for results without lake. Second, the binary metrics POD, FAR, and CSI are closely related to CC and relatively insensitive to high flow magnitude, while NSE is highly favoured by high flow. So, the binary metrics offer an alternative yet important view to flood forecasting because the timing to detect a flood event is essential. Third, the intention of this framework is to develop an operational flood forecast system that weather forecasters use to issue warnings. A map of gauges with FAR could help us identify regions that model does not perform well, which is insightful for forecasters and decision makers.

We want to gently remind the reviewers/readers on the implication of a negative NSE, which suggests systematic bias but not necessarily poor CC. For a categorical detection of flood events (as indicated by the binary metrices), CC is as important as, if not more important than, bias. Therefore, NSE alone is not sufficient for making an informative judgement on the simulation performance, which is why we try to provide more metrics here.

- Supplementary materials should only be provided to corroborate existing evidence in the paper; the authors use Fig S1 to make a completely distinct point. This content should be either removed or incorporated into the main document.

Response:

Thanks for your suggestions. The supplementary Figure is now moved to the main text.

- The authors have multiple discussions of ideas that are very loosely related to the paper and/or not tied to any specific results herein, and should be removed:
    1. Ensemble forecasts at line 229
    2. Advocating for modular model structure at line 377
    3. Support for parameter regionalization line 388
    4. Two-way feedback between social systems and catchment signatures (ln 398)
    5. Future work on machine learning based reservoir operation simulation (line 402)

Response:

Thanks for your suggestions. We removed those discussion points mentioned above and expanded points (limitations and discoveries) that related to this study.

- The authors use percent change in NSE and BIAS to present results (e.g., in figure 6). However, this metric is very problematic for variables that can be positive or negative, because the denominator can go to zero. Another metric must be used.

Response:

Thanks for your comments. We changed the percent change to absolute change as attached below to replace the original Fig. 6.

[Figure]

- Multiple minor issues
    6. Bias is reported both as a percentage 0-100% and as a floating point 0-1 (fig 5)

Response: Thanks for your comments. We unified Bias as floating point throughout our main text.

    7. Speedup is usually used to evaluate improvements in computational costs; the authors here only report (less generalizable) differences in raw run times per time step.

Response:

The speedup is taken as the average run times per time step over the whole simulation period. We added one more row to compare the total computational costs.

**Table 1. Statistical comparison of model performance over the continental U.S. Bolded numbers indicate the best metrics among the three model configurations. The computational speed is calculated as an average speed over a whole simulation period.**

| Metrics | Gridded CREST (Flamig et al., 2020) | CREST-VEC (w/o lake) | CREST-VEC (w/ lake) |
|---|---|---|---|
| Simulation resolution | 1 km | **90 m** | **90 m** |
| Total computational cost (hours) | 149.2 | **29.9** | 32.96 |
| Computational Speed for routing (sec/step) | 7.2 | **0.35** | 0.37 |
| Max NSE | 0.71 | **0.87** | **0.87** |
| Median NSE | -0.06 | 0.12 | **0.18** |
| % gauges NSE>0 | 41.8 % | 50.6 % | **56.2 %** |
| Max CC | **1.0** | 0.96 | 0.96 |
| Median CC | 0.40 | **0.67** | **0.67** |
| Median bias | **9%** | 27% | 17% |

8. The reasons provided for changing bias at line 247 are implausible – to influence bias you need to have water leave the domain by means other than streamflow. A more likely scenario is that this would be due to evaporation from the lake surface.

Response:

Thanks for your comments. Here the reduction of bias is measured at stream gauges, so if water is held in upstream lakes, streamflow measured at downstream gauges is much lower than simulation without lake. It is the primary (most obvious) reason for reducing the bias. Of course, there are other factors like lake evaporation or water abstraction, but those factors have so far not been considered in our continental simulation.

9. D-infinity (line 78) is also a grid-based (not vector-based) algorithm, and should be cited as Tarboton (1997)

Response:

Thanks for your suggestions. Yes, Dinf is a raster-based approach. Here we wanted to mention the vector-based routing can allow water infow or outflow from any direction. To avoid confusion, we deleted this sentence. For the reference you mentioned, we inserted in previous raster-based routing.

10. The CREST model original paper (Wang 2011) should be included upon its first mention. How is this paper not referenced?

Response:

Thanks for your comments. It was cited in original line L.129.

11. Many minor text errors not inventoried here given expectation of significant revisions

Response:

Thanks for your comments. After revisions, we carefully scrubbed the main text.

References

Han, M., J. Mai, B.A. Tolson, J.R. Craig, É. Gaborit, H. Liu, and K. Lee, Subwatershed-based lake and river routing products for hydrologic and land surface models applied over Canada, Canadian Water Resources Journal, doi:10.1080/07011784.2020.1772116, 2020

Tarboton, D. G., (1997), "A New Method for the Determination of Flow Directions and Contributing Areas in Grid Digital Elevation Models," Water Resources Research, 33(2): 309-319

Jiahu Wang, Yang Hong , Li Li , Jonathan J. Gourley , Sadiq I. Khan , Koray K. Yilmaz , Robert F. Adler , Frederick S. Policelli , Shahid Habib , Daniel Irwn , Ashutosh S. Limaye , Tesfaye Korme & Lawrence Okello (2011) The coupled routing and excess storage (CREST) distributed hydrological model, Hydrological Sciences Journal, 56:1, 84-98, DOI: 10.1080/02626667.2010.543087

---

## Author Response (AR2)

The authors should address the below minor comments generated by Reviewer #2.

1. Figure 10: unequal space between vertical subplots.

Response: Thanks for your comments. Figure has been re-generated with uniform space.

2. Figure 8: the main categories (i.e. Soil, Hydrology and Climate) can be placed as the y-label for clarity and the parameters should be in their full name.

Response: Thanks for your comments. Figure 8 has been deleted during our first revision according to other reviewers' suggestions.

3. Figure 5: The numbers are cut off. Caption need to be rewritten formally.

Response: Thanks for your comments. Figure 5 has been regenerated and Figure Caption has been rewritten.

4. Figure 4: Event Harvey should be changed to Hurricane Harvey

Response: Thanks for your suggestions. We changed text Event Harvey to Hurricane Harvey in this iteration.

5. The authors should be consistent in map projection. The US map is plotted in different projections which may hinder the interpretation of the results.

Response: Thanks for your comments. We affirm that all the map projection is in US Lambert Conformal projection.